***Nat Microbiol.* Author manuscript; available in PMC 2021 December 03.**

# Global phylogeny of *Treponema pallidum* lineages reveals recent expansion and spread of contemporary syphilis

**Mathew A. Beale**[1,*], **Michael Marks**[2,3], **Michelle J. Cole**[4], **Min-Kuang Lee**[5], **Rachel Pitt**[4], **Christopher Ruis**[6,7], **Eszter Balla**[8], **Tania Crucitti**[9], **Michael Ewens**[10], **Candela Fernández-Naval**[11], **Anna Grankvist**[12], **Malcolm Guiver**[13], **Chris R. Kenyon**[9], **Rafil Khairullin**[14], **Ranmini Kularatne**[15], **Maider Arando**[16], **Barbara J. Molini**[17], **Andrey Obukhov**[18], **Emma E. Page**[19], **Fruzsina Petrovay**[8], **Cornelis Rietmeijer**[20], **Dominic Rowley**[21], **Sandy Shokoples**[22], **Erasmus Smit**[23,24], **Emma L. Sweeney**[25], **George Taiaroa**[26], **Jaime H. Vera**[27], **Christine Wennerås**[12,28], **David M. Whiley**[25,29], **Deborah A. Williamson**[26], **Gwenda Hughes**[30], **Prenilla Naidu**[31,22], **Magnus Unemo**[32], **Mel Krajden**[5,33], **Sheila A. Lukehart**[34], **Muhammad G. Morshed**[5,33], **Helen Fifer**[35], **Nicholas R. Thomson**[1,2,*]

[1]Parasites and Microbes Programme, Wellcome Sanger Institute, Hinxton, United Kingdom

[2]Faculty of Infectious and Tropical Diseases, London School of Hygiene & Tropical Medicine, United Kingdom

[3]Hospital for Tropical Diseases, University College London Hospitals NHS Foundation Trust, London, United Kingdom

[4]HCAI, Fungal, AMR, AMU and Sepsis Division, UK Health Security Agency, London, United Kingdom

[5]British Columbia Centre for Disease Control, Public Health Laboratory, Vancouver, British Columbia, Canada

[6]Molecular Immunity Unit, University of Cambridge Department of Medicine, MRC-Laboratory of Molecular Biology, Cambridge, United Kingdom

[7]Department of Veterinary Medicine, University of Cambridge, Cambridge, United Kingdom

[8]Bacterial STIs Reference Laboratory, Department of Bacteriology, National Public Health Centre, Budapest, Hungary

[9]Department of Clinical Sciences, Institute of Tropical Medicine, Antwerpen, Belgium

[10]Brotherton Wing Clinic, Brotherton Wing, Leeds General Infirmary, Leeds, United Kingdom

*Correspondence to mathew.beale@sanger.ac.uk or nrt@sanger.ac.uk.

Author Contributions Statement
Conceived and designed the study: MAB, MM, SAL, NRT. Coordinated collaboration, receipt and sequencing of samples: MM, MAB, MU. Collected, retrieved and prepared samples and patient metadata: MM, MJC, M-KL, RP, EB, TC, ME, CFN, AG, MG, CRK, RKh, RKu, MA, BJM, AO, EP, FP, CRi, DR, SS, ESm, ELS, GT, JHV, CW, DMW, DAW, GH, PN, MK, MU, SAL, MGM, HF. Performed laboratory work for sequencing: MAB, GT. Analysed the data: MAB. Provided analytical tools and advice: CRu. Wrote the initial draft of the paper: MAB, with substantial contributions from NRT. All authors viewed and contributed to the final paper.

Competing Interests Statement
MK declares institutional funding from Roche, Hologic and Siemens unrelated to this work. The remaining authors have no competing interests to declare. The funders had no input into the study design, interpretation or decision to submit for publication.

[11]Microbiology Department, Vall d'Hebron Research Institute, Universitat Autònoma de Barcelona, Barcelona, Spain

[12]National Reference Laboratory for STIs, Department of Clinical Microbiology, Sahlgrenska University Hospital, Gothenburg, Sweden

[13]Laboratory Network, Manchester, UK Health Security Agency, Manchester Royal Infirmary, Manchester, United Kingdom

[14]Institute of Fundamental Medicine and Biology, Kazan Federal University, Kazan, Russia

[15]Centre for HIV & STI, National Institute for Communicable Diseases, Johannesburg, South Africa

[16]STI Unit Vall d'Hebron-Drassanes, Infectious Diseases Department, Hospital Vall d'Hebron, Barcelona, Spain

[17]Department of Medicine, University of Washington, USA

[18]Tuvan Republican Skin and Venereal Diseases Dispensary, Ministry of Health of Tuva Republic, Kyzyl, Tuva Republic, Russia

[19]Virology Department, Old Medical School, Leeds Teaching Hospitals Trust, Leeds, United Kingdom

[20]Colorado School of Public Health, University of Colorado, Denver, Colorado, USA

[21]Midlands Regional Hospital Portlaoise, Co. Laois, Ireland

[22]Alberta Precision Laboratories, Edmonton, Canada

[23]Clinical Microbiology Department, Queen Elizabeth Hospital, Birmingham, United Kingdom

[24]Institute of Environmental Science and Research, Wellington, New Zealand

[25]The University of Queensland Centre for Clinical Research, Faculty of Medicine, The University of Queensland, Brisbane, Queensland, Australia

[26]Department of Microbiology and Immunology at The Peter Doherty Institute for Infection and Immunity, Melbourne, Australia

[27]Department of Global Health and Infection, Brighton and Sussex Medical School, University of Sussex, United Kingdom

[28]Department of Infectious Diseases, Institute of Biomedicine, University of Gothenburg, Gothenburg, Sweden

[29]Pathology Queensland Central Laboratory, Queensland, Australia

[30]Department of Infectious Disease Epidemiology, London School of Hygiene & Tropical Medicine, London, United Kingdom

[31]Department of Laboratory Medicine and Pathology, Faculty of Medicine, University of Alberta, Alberta, Canada

[32]WHO Collaborating Centre for Gonorrhoea and other Sexually Transmitted Infections, National Reference Laboratory for STIs, Faculty of Medicine and Health, Örebro University, Örebro, Sweden

[33]Department of Pathology and Laboratory Medicine, University of British Columbia, Vancouver, British Columbia, Canada

[34]Departments of Medicine/Infectious Diseases & Global Health, University of Washington, USA

[35]Blood Safety, Hepatitis, STI & HIV Division, UK Health Security Agency, London, United Kingdom

## Abstract

Syphilis, which is caused by the sexually-transmitted bacterium *Treponema pallidum* subspecies *pallidum*, has an estimated 6.3 million cases worldwide per annum. In the past ten years, the incidence of syphilis has increased by more than 150% in some high-income countries, but the evolution and epidemiology of the epidemic are poorly understood. In order to characterize the global population structure of *T. pallidum* we assembled a geographically and temporally diverse collection of 726 genomes from 626 clinical and 100 laboratory samples collected in 23 countries. We applied phylogenetic analyses and clustering and found that the global syphilis population comprises just two deeply branching lineages, Nichols and SS14. Both lineages are currently circulating in 12 of the 23 countries sampled. We subdivided *Treponema pallidum* subspecies *pallidum* into 17 distinct sublineages to provide further phylodynamic resolution. Importantly, two Nichols sublineages have expanded clonally across 9 countries contemporaneously with SS14. Moreover, pairwise genome analyses revealed examples of isolates collected within the last 20 years from 14 different countries that had genetically identical core genomes, which might indicate frequent exchange through international transmission. It is striking that the majority of samples collected before 1983 are phylogenetically distinct from more recently isolated sublineages. Using Bayesian temporal analysis we detected a population bottleneck occurring during the late 1990s followed by rapid population expansion in the 2000s that was driven by the dominant *T. pallidum* sublineages circulating today. This expansion may be linked to changing epidemiology, immune evasion, or fitness under antimicrobial selection pressure, since many of the contemporary syphilis lineages we have characterised are resistant to macrolides.

Syphilis, caused by the bacterium *Treponema pallidum* subspecies *pallidum* (TPA), is a prevalent sexually transmitted infection which can cause severe long-term sequelae when left untreated. Historically, syphilis is commonly believed to have caused a large epidemic across Renaissance Europe, having previously been absent or unrecognised[1,2]. Although the origins of syphilis, and the accurate dating of the most recent common ancestor of TPA are still the subject of debate[3–6], it is suggested that the strains of TPA that persist in human populations to this day can be traced back to that introduction into Western Europe approximately 500 years ago, and subsequently disseminated globally[3,4,6].

Following the introduction of effective antibiotics after World War II, syphilis incidence fluctuated[7] without disappearing, until the 1980s and 1990s during the HIV/AIDS crisis when disease incidence declined markedly[8], linked to community wide changes in

sexual behaviour, shifting of affected populations, AIDS-related mortality and widespread antimicrobial prophylaxis of HIV-infected populations. However, since the beginning of the 21st century, there has been a substantial resurgence in syphilis incidence[9–14]. In many countries, this has been associated with populations of men who have sex with men (MSM) engaging in high risk sexual activity[11,15]. Transmission between MSM and heterosexuals is a particular concern due to the risk of *in utero* transmission to the foetus, leading to congenital syphilis[16].

Previous genomic analyses of TPA genomes have described two deep branching phylogenetic lineages 'SS14' and 'Nichols'[3]. SS14-lineage strains represent the vast majority of published genomes[4], and phylogenetic analysis showed that the origins of the SS14-lineage can be traced back to the 1950s[3], followed by subsequent expansion of sublineages occurring during the 1990s[4]. Our understanding of the Nichols-lineage is predominantly limited to laboratory strains from the USA, with relatively few clinical strains sequenced[4,17]. However, most TPA genomes published to date originate from the USA[4], Western Europe[3,4,17,18] and China[19,20], and our understanding of the true breadth of diversity of TPA is incomplete[21]. Our view of the diversity of syphilis samples predating the 21st century is even more limited, and these issues are partly explained by the fact it has not been possible to culture TPA outside of a rabbit until recently[22]..

In this multicentre collaborative study, we used direct whole genome sequencing to generate a global view of contemporary syphilis from patients in Africa, Asia, the Caribbean, South America and Australia sampled between 1951 and 2019. Our dataset also includes a detailed analysis of the 'within-country' variation seen in TPA genomes in North America and Europe. We present evidence of globally spanning transmission networks with identical strains found in dispersed countries, indicating that, based on our data, TPA is genetically homogenous. Furthermore, we show that this genetic homogeneity is the result of a rapid and global expansion of TPA sublineages occurring within the last 30 years following a population bottleneck. This means the TPA population infecting patients in the 21st century is not the same as that infecting patients in the 20th.

## Describing the global population structure of *Treponema pallidum* subspecies *pallidum*

We performed targeted sequence-capture whole genome sequencing on residual genomic DNA extracted from diagnostic swabs taken from TPA PCR-positive syphilis patients, and on TPA strains previously isolated in rabbits. We combined these data with 133 previously published genomes[3,4,18–20,23–26]. After assessment for genome coverage and quality, we had a total of 726 genomes with >25% of genome positions at >5X coverage (mean 82%, range 25-97%), sufficient for primary lineage classification. This dataset included 577 new genomes sequenced directly from clinical samples, and 16 new genomes sequenced from samples passaged in rabbits.

Our dataset includes 23 countries (range 1-355 genomes per country; Figure 1A, 1C), including previously poorly or unsampled regions including Africa (South Africa (n=1) and Zimbabwe (n=18)), Scandinavia (Sweden, n=7), Central Europe (Hungary, n=20), Central

Asia (Tuva Republic, Russia, n=10) and Australia (n=5), as well as substantially increasing the sampling from North America (Canada, n=157; United States, n=86) and Western Europe (Spain, n=5; Belgium, n=1; Ireland, n=4; the United Kingdom, n=355). We also include previously published genomes from South America (Argentina, n=1), the Caribbean (Cuba, n=3; Haiti, n=1), and elsewhere. Due to a lack of long term archived samples, 96.0% (n=697) of samples were collected from 2000 onwards (Figure 1B, 1E). Samples collected prior to 2000 (n=29) were passaged in the *in vivo* rabbit model (Supplementary Data 1), whereas most samples collected from 2000 onwards (89.8%, 626/697) were sequenced directly from clinical samples.

Phylogenetic analysis assigned all genomes into one of two deeply branching lineages ("Nichols" or "SS14") (Extended Figure 1). Looking across all well sampled countries (Figure 1C, 1D), from the first detection of the modern SS14-lineage (excluding the outlying 1953 Mexico A strain) in the 1970s, we consistently see both lineages circulating broadly through until 2019 (Figure 1B, 1E). More specifically 81.3% (n=590) of genomes belonged to the SS14-lineage and 18.7% (n=136) to the Nichols-lineage, and in the 12/23 countries where both lineages were present, 80.3% (544/677, median per country 75.3%, range 33.3-93.3%) were SS14-lineage (Figure 1D, 1F). In the 11 countries showing only a single lineage, most had three or fewer samples, the notable exceptions being Portugal (n=19), Sweden (n=7) and Russia (n=10) (Figure 1C).

## Fine scaled analysis of SS14- and Nichols-lineage phylogenies

To answer finer scaled evolutionary questions, we filtered our dataset to focus on the 528 genomes with >75% genome sites at >5X (genome length 1,139,569 bp, mean % sites 92.9%, range 75.1–96.9%) and a mean coverage of 111X (range 11X–727X). This filtered dataset comprised 401 new and 127 published genomes (Supplementary Data 1), but excluded the only sample from Belgium, leaving 22 countries in the analysis. After excluding 19 regions of recombination and genomic uncertainty due to gene paralogy or repetitive regions[3,4,6], we used Gubbins[27] to infer a further 19 regions of putative recombination (see Methods and Supplementary Data 2 for details). We refer to the resulting masked sequence alignment as the core genome and used it to infer a whole genome maximum likelihood phylogeny using IQ-Tree[28] (Supplementary Figure 1). To define sublineage clusters we used 100 bootstrapped trees as independent inputs to rPinecone[29] with a 10 SNP threshold as previously[4], and evaluated their consistency using hierarchical clustering (Supplementary Figure 2). We found broad support for the Nichols sublineages across all conditions evaluated, but some parts of the SS14-lineage phylogeny were less well supported. To focus on the more stable sublineages, we required that at least 5% of the bootstrap replicates supported a cluster (the most conservative threshold tested – see Supplementary Figure 2 and Methods). Using this approach, we defined 17 sublineages and 8 singleton strains across both SS14- and Nichols-lineages (SS14: 426 genomes divided into 5 sublineages and 4 singletons; Nichols: 102 genomes divided into 12 sublineages and 4 singletons; Figure 2; Supplementary Figure 1, Extended Figure 4, Extended Figure 5). Sublineage 6 diverged from other genomes very close to the common ancestor of TPA, and due to low total diversity in the dataset can appear on either side of the root (Nichols or SS14) depending on the phylogenetic approach used; for the purposes of this analysis, we

classified sublineage 6 strains as Nichols-lineage, since they are more distantly related to the recent contemporary SS14-lineage expansions.

From Figure 2 it is apparent that the phylogeny of SS14-lineage is dominated by SS14 sublineage 1 (n=365), comprised of closely related genomes present in 18 countries and six continental groupings (Asia, Caribbean, Europe, North America, Oceania, and South America). The oldest example of sublineage 1 was collected in 1981 (TPA_USL-SEA-81-3, Seattle USA), and the most recent samples were from 2019 (Figure 2C). Sublineage 2 (n=32; Figure 2E, Extended Figure 2) contained samples from Canada, China, the UK and the USA. In a previous analysis[4], we manually divided this sublineage into two groups (one from China, one from the USA), based on temporal and geospatial divergence, and the independent evolution of different macrolide resistance alleles. However, by adding new genomes (Extended Figure 2), we now see that even the original cluster of samples from China is interspersed with genomes from the UK (n=1) and Canada (n=4), indicating that this is not a geographically restricted group.

The 12 2015 Zimbabwean genomes in our study formed two distinct clades, one nested within SS14-lineage (sublineage 4, n=8, also including a distantly related singleton from the USA 1981, TPA_USL-SEA-81-8), the other within Nichols-lineage (sublineage 13, n=4) and exclusively found in Zimbabwe. We also examined 10 genomes collected in the Tuva Republic, central Russia in 2013/2014, and these were distributed between three different SS14 sublineages (1, 3, 5). Sublineage 5 was found only in Tuva, whilst sublineage 3 was found throughout Europe (Czech Republic, Hungary, Sweden and the UK; Figure 2E, Extended Figure 2), with the remaining sample from Russia belonging to the highly expanded global SS14 sublineage 1.

Consistent with previous studies[3,4,17] Nichols-lineage strains were genetically more diverse, with longer branch lengths and higher nucleotide diversity than SS14-lineage (Nichols $\pi=3.2\times10^{-5}$, SS14 $\pi=6\times10^{-6}$), reflecting the predicted time of lineage diversification. However, our increased sampling also revealed two recent clonal expansions within the Nichols-lineage: sublineage 14 (n=55), comprising samples from Canada, Hungary, Spain and the UK (Figure 2E) and sublineage 8 (n=15) comprising samples from the Australia, Canada, France, Ireland, the Netherlands, the UK, and the USA (Extended Figure 3).

In addition to observing evidence of recent clonal expansions we also show greater resolution of Nichols-lineage diversity, identifying two new samples from UK patients (PHE130048A, PHE160283A, collected in 2013 and 2016 respectively) which occupy positions basal to all Nichols-lineage strains (Extended Figure 3). Indeed, this analysis suggested the most recent common ancestor of this sublineage was very close to the root of all TPA.

Multiple derivatives of the highly passaged prototype Nichols-lineage strain (denoted Nichols-1912) isolated in 1912 were included in our analysis (Nichols_v2, Seattle_Nichols, Nichols_Houston_E, Nichols_Houston_J, and Nichols_Houston_O). Figure 2B (Extended Figure 4) shows that derivatives of Nichols-1912 fall within a distinct clade which also includes independently collected contemporaneous clinical samples (some with minimal

passage through the rabbit model), including a sample (TPA_AUSMELT-1) collected in Australia in 1977[30]. This clade could be subdivided into four sublineages (9, 10, 11, 12) and one singleton. Notably, the last sample belonging to this clade was collected in 1987 (TPA_USL-Phil-3). Hence, within our sampling framework this appears to be an example of a historic lineage declining to rarity or even becoming extinct. More broadly, we note that although this cluster of both clinical and laboratory strains were all passaged to varying degrees through the rabbit, other samples passaged in the rabbit were distributed throughout the phylogeny and were present in 9/17 sublineages (Extended Figure 5).

## Temporal analysis of population dispersal

To infer temporal patterns within the global phylogeny, we performed Bayesian phylogenetic reconstruction using BEAST[31] under a Strict Clock model with a Bayesian Skyline population distribution. We excluded 8 samples from strains known to be heavily passaged or with uncertain collection dates from the previous dataset of 528, giving a dataset of 520 samples and 883 variable sites. We inferred a median molecular clock rate of $1.28 \times 10^{-7}$ substitutions/site/year (95% Highest Posterior Density (HPD) $1.07 \times 10^{-7}$ – $1.48 \times 10^{-7}$), equivalent to one substitution/genome every 6.9 years, consistent with recent analyses[4,6].

Within the global TPA phylogeny (Figure 3A), we observed several patterns of genomic dispersal. The first reflects the separation of the Nichols- and SS14-lineages (median date in our analysis 1534, 95% HPD 1430-1621), the subject of much previous analysis[3,4,6]. These data also showed that the common ancestor of these lineages is separated from recent samples by long phylogenetic branches and an absence of older ancestral nodes, suggesting unsampled historical diversity, and that most contemporary TPA descend from much more recent ancestral nodes. We previously dated the common ancestors for clonal expansions of 9 SS14 sublineages between the 1980s and early 2000s[4]. With this expanded dataset, we focussed on the major clonally expanded sublineages 1, 2, 8 and 14, each having at least 15 samples.

Next, we used Bayesian Skyline analysis to determine the relative genetic diversity over different time periods in the phylogeny (Figure 3B), showing a very sharp decline during the 1990s and 2000s, followed by an equally sharp rise that continued until present day. To test the statistical support for this expansion, we extracted the proportion of trees in the posterior distribution supporting a >2-fold population expansion above the population mean (68.6%, 9263/13503 trees), and plotted the distribution of expansion start dates (Figure 3C) (median date 2011). We further tested the proportion of trees supporting a 2-fold population decline between 1990-2015 (90.7%, 12248/13503 trees, median date 2000) and a 2-fold population expansion between 1990-2015 (59.0%, 7966/13503 trees, median date 2012) (Supplementary Figure 3). These findings were also apparent in SS14 sublineages 1 and 2 (Extended Figure 6) but not for Nichols sublineage 8. We had insufficient temporal signal to repeat this analysis on multi-country expanded Nichols sublineage 14 (Supplementary Figure 4). We independently analysed SS14- and Nichols-lineages, and this indicated that the population decline was largely associated with Nichols-lineage, and coincided with expansion of SS14-lineage (Extended Figure 7A). However, our analysis also shows that the

Nichols-lineage continued to diversify after 2010 (Extended Figure 7B), consistent with our analysis of clonally expanded Nichols sublineages.

## Global sharing of sublineages and identical strains

To further understand the patterns of recent population expansion we sought evidence of sharing of sublineages among countries, classifying sublineages as singletons (n=8), private to a country (n=8), or multi-country (n=9), and found that 20/22 countries contained at least one multi-country sublineage (Figure 4A, Supplementary Figure 5). We inferred pairwise SNP distances for genomes within and between each country (Figure 4C, 4D), and where there was more than one sample (n=18), we found fewer than 26 (SS14-lineage) and 80 (Nichols-lineage) pairwise SNPs separating genomes within any one country (Figure 4C), illustrating the close genetic relationships between samples (particularly SS14-lineage). We also found very low genetic distance between paired samples from different countries, with 27 country pairings (14 countries) showing zero core genome pairwise SNPs (Figure 4D). In particular Canada, UK and USA, with the highest sampling (Figure 4B), showed the most zero pairwise interactions with other countries (Figure 4D). Therefore, we cannot rule out similar transmission events occurring between other countries. We compared pairwise SNP distances with geographical distance between country centroids. Although we found a moderate correlation for Nichols-lineage (Pearson's correlation 0.49, p<0.001, 9620 comparisons), this was lower for SS14-lineage (0.31, p<0.001, 181476 comparisons) and for the four largest multi-country sublineages (sublineage 1, 0.09, p<0.001, 133225 comparisons; sublineage 2, 0.43, p<0.001, 1024 comparisons; sublineage 8, 0.27, p<0.001, 225 comparisons; sublineage 14, 0.08, p<0.001, 3025 comparisons) (Extended Figure 8). Hence, overall this indicates weak geographical structure for TPA.

To understand these observations more fully, we focussed on British Columbia (Canada; BC) and England, both of which have experienced a recent rise in syphilis incidence (Extended Figure 9A), and for which we had a large number of samples. Included were 84 high quality BC genomes collected by the BC Centre for Disease Control between 2000 and 2018. From England, we had 240 high quality genomes from samples taken by the National Reference Laboratory at Public Health England (n=198) and four non-referring laboratories (n=42) collected between 2012 and 2018. In BC, SS14 sublineage 1 dominated throughout the 18-year survey period, representing 82% of all BC genomes (Extended Figure 9B). In addition, isolated cases of SS14 sublineage 2 were seen in 2000 and 2012 as well as a single Nichols-lineage sample (singleton) in 2002 (Extended Figure 9B). Then from 2013 onwards, we detected two new Nichols sublineages: Nichols sublineage 8 and sublineage 14. The latter two lineages were also found across USA and Europe (Figure 2E).

Both Nichols- and SS14-lineages were consistently present in the English samples between 2012 and 2018. All of the common sublineages (4/4) found in BC were also present in England, as well as 4 additional sublineages (Nichols sublineages 6, 15, 16; SS14 sublineage 3) and one SS14 singleton strain not detected in BC (Extended Figure 9B). Sublineage 14, first detected in BC in 2013, was also the most numerous Nichols sublineage in England, but was not detected here until 2014.

Pairwise SNP distances between orthogonal genomes from the same sublineage showed 2622 pairwise combinations of BC (n=56) and English samples (n=78) sharing zero pairwise SNPs over the core genome alignment for isolates collected between 2004 and 2019. To understand the effect of temporal distance we compared both the pairwise SNP distance and the pairwise time distance (in years) between genomes from the same sublineage (Extended Figure 9D). These data showed that the mean number of years separating identical core genomes was 2.5 years (range 0-15), and the mean temporal distances of identical genomes were similar within BC (2.9 years) and England (1.9 years) and between the two (2.7 years). The number of pairwise SNPs increased with temporal separation across all BC and English genomes from the same sublineage (Pearson's Correlation 0.126, p<0.001, 55841 comparisons), with a mean of 4.9 SNPs (range 0–23) separating genomes from the same year and sublineage, compared to 7.8 SNPs (range 6-11) after 19 years (Extended Figure 9D). This means that inference of direct patient-to-patient TPA transmission using the core genome will be challenging at the population level, and limit opportunities for real-time genomic epidemiology because identical genomes can be separated by many years, and confidence intervals around temporal reconstructions will be broad. In the case of sublineage 14, we first detected this in BC, then the following year in England. Since we had a deeply sampled survey of populations over time for both countries, it seems plausible that this represents a novel introduction into BC and England. However, low temporal rates and incomplete sampling, mean this must be interpreted with caution.

We also found some rarer sublineages – either as singleton strains, or those private to a single country. Whilst this might be expected in poorly sampled and geographically distant locations, such as Cuba, Haiti, Mexico and Zimbabwe, we found that the majority of private (6/8) and singleton (5/8) sublineages were actually from Canada, the UK or USA (Figure 4A), suggesting deeper sampling elsewhere will also find novel diversity.

Given our observations of individual sublineage expansion, we investigated whether the expansion could be related to antimicrobial resistance. Overlaying SNPs known to confer macrolide resistance (A2058G, A2059G) in the ribosomal 23S rRNA gene on the phylogeny showed evidence of macrolide resistance in 6/9 multi-country sublineage expansions (Extended Figure 10), with the majority of samples being resistant in the largest sublineages 1, 2, 8 and 14. In contrast, only one private sublineage (sublineage 6, n=2) contained a macrolide resistant sample, suggesting that macrolide resistance is potentially linked to expansion in multicountry sublineages.

We further explored the genomic differences between SS14- and Nichols-lineages in our core genome alignment, using ancestral reconstruction to infer the common ancestral sequences of contemporary SS14- and Nichols-lineages (Supplementary Figure 6). We identified 95 SNPs separating these clades, and functional annotation indicated 16 SNPs in 11 genes resulted in non-synonymous coding changes, 59 SNPs were synonymous, and 20 were intragenic (see Supplementary Data 3). We noted a high number of non-synonymous changes to the *bamA* gene (TPASS_RS01600; involved in outer membrane protein assembly), but none of these genes is known to confer a fitness advantage that may distinguish the two lineages.

Previous attempts to understand the origins of the original syphilis pandemic[3,6,21], as well as the dynamics of the current one[3,4], have been constrained by the technical difficulty of sequencing TPA genomes, as well as relatively small datasets, with limited geographical diversity and sampling biases. In our study, we assembled the most comprehensive collection of syphilis samples from around the world to date, including samples from both the 20th and 21st century. Despite this, we still find that the TPA population consists of just two deep branching lineages, SS14-lineage and Nichols-lineage, with no outlying lineages. We were able to show that these lineages are both globally distributed and, where we have densely sampled, we find the relative proportions of each to be consistent. Although the overall diversity detected within the Nichols-lineage is far greater than that of SS14-lineage, suggesting earlier dissemination, we also found that these two major lineages exhibit similar phylodynamics, with recent sublineage expansions apparent in both lineages. This suggests that both Nichols- and SS14-lineages are capable of exploiting the transmission pathways driving the current syphilis epidemic.

Amongst our data, we sequenced the first genomes from syphilis patients in Africa, and our analysis shows that these genomes represented novel private sublineages, but their genomic diversity is nested entirely within the existing phylogenetic framework – these TPA genomes are not unusual. Indeed, we even observed the same pattern of Nichols- and SS14-lineages both being present in Zimbabwe, suggesting multiple introduction events into Zimbabwe. The same was true for genomes sequenced from Central Russia, where the private sublineage 5 represented novel, but entirely unremarkable genomic diversity.

In our study, we found that sublineages and genetically similar genomes were more likely to be shared among deeply sampled countries. This suggests that sublineage sharing between countries is high, and deeper sampling of other countries will likely reveal similar patterns of sharing. As sampling depth increased we also found rarer sublineages, notably sublineage 6, representing novel outlying genetic diversity basal to all contemporary Nichols-lineage examples, in 2/240 contemporary UK patients. This suggests that some sublineages may truly be rare, whilst the high frequency of other sublineages could reflect either fitness advantages or epidemiological factors such as infecting patients within particularly high-risk sexual networks, allowing these sublineages to expand more successfully. Singleton or private sublineages could reflect insufficient sampling of a country or region, or sampling biases within a country These sublineages may therefore reflect transmission networks that are either contained within a less internationally mobile demographic, or may reflect transmission networks common in a region that is otherwise poorly sampled (e.g. Africa).

Our observation that the well-studied Nichols reference genomes (largely derived from or related to the original Nichols-1912 isolate) form an isolated clade, not represented in contemporary TPA, is important. One possible explanation is that these samples form a distinct clade due to convergent evolution in the rabbit model. However, this clade contains samples both extensively and minimally passaged, whilst other samples passaged in rabbits are distributed throughout the broader phylogeny, included in three SS14 sublineages (1, 2, 4) and two additional Nichols sublineages (7, 8). This indicates that passage in the rabbit model has not biased other parts of the phylogeny. The majority of Nichols-lineage strains collected prior to 1988 belong to this clade, and these samples mostly come from

a small group of laboratories in the USA. Therefore, it is also tempting to suggest that this reflects a sampling bias for that time period. However, the phylogenetic placement of TPA_AUSMELT-1 within the same clade, isolated in 1977 in Australia[30], and independently cultured and sequenced, contradicts this hypothesis, and may suggest that this clade represents the dominant TPA of the period. The complete absence of related genomes in contemporary sampling could represent a decline to becoming a rare or even extinct lineage and therefore implies that the Nichols reference strain is not representative of contemporary syphilis, or even contemporary Nichols-lineage strains.

Our data show that for some sublineages, modern syphilis is a truly global disease, with shared lineages, sublineages and indeed nearly identical strains found all over the world. The large expansions of highly related genomes, in particular sublineage 1, represent the bulk of our dataset, and the widespread sharing of major lineages suggests we have sampled from a series of globally contiguous sexual networks.

Furthermore, we find evidence of a striking change in the genetic diversity and effective population size of TPA genomes, suggestive of a possible population bottleneck occurring between the late 1990s and early 2000s. This was followed by a rapid expansion of certain sublineages, leading to the contemporary TPA population structure. This bottleneck, potentially a consequence of post-HIV safe-sex messaging, persistent antimicrobial prophylaxis in at risk HIV-positive populations, and possibly HIV-associated mortality, appears to have led to a striking duality in the TPA dominating populations before and after. The rapid expansion may be attributed to a relaxation of sexual behaviour following the widespread introduction of highly active antiretroviral therapy. Notably, although macrolide resistance was not universally distributed throughout the phylogeny or present in all sublineages, most of the multi-country sublineages were largely macrolide resistant, and this could also have played a role through off-target effects, e.g. during treatment of other (particularly sexually transmitted) infections[32]. Azithromycin and other macrolides are no longer recommended treatments at any stage of syphilis in the European syphilis management guidelines[33]. A further possibility is that partial host immunity plays a role in sublineage dynamics, suggested by previous modelling of aggregated national surveillance data[34] and reported attenuated symptoms on reinfection[35,36], and this will require detailed investigation of well characterised sexual networks, and examination of hypervariable antigenic genes excluded from the core genome used in our phylogenetic analysis.

Our study has a number of limitations. Our samples were collected in an opportunistic manner, using residual samples available in regional or national archives. Since our sampling coverage is uneven, with some countries either missing or under-sampled, it was not possible to infer the direction of transmission between countries. Whilst sampling was particularly limited in Africa, Asia and South America, we still provide a snapshot of strains from these regions, all of which overlap with the genetic diversity of our more deeply sampled regions (North America and Europe), suggesting we have captured dominant global lineages. We also show that even deeply sampled countries can harbour rare sublineages, and it is therefore likely that future studies will reveal further novel diversity. Samples collected prior to the widespread adoption of molecular diagnostics in the early 2000s are limited here, and this is largely influenced by the difficulty in isolating new strains prior to the recent

development of *in vitro* culture[22], and the lack of long-term storage of clinical swabs. Most (but not all) older strains come from the USA, and this could mean that we do not accurately reflect the global population structure prior to 2000. Moreover, because of limited sampling prior to 2000, historic lineages that lack extant descendants (e.g. due to extinction) would not be modelled by our Bayesian Skyline, and this could limit our estimates of historic population diversity.

Despite this our data show the *T. pallidum* infecting patients today is not the same *T. pallidum* infecting patients even 30 years ago – ancestral sublineages may have become extinct, being replaced by new sublineages that have swept to dominance across the globe with the dramatic upswing in syphilis cases in the US, UK, and other Western European countries, which were heavily sampled in our study. That such a bottleneck is linked to HIV-related behavioural change during the 1990s, rather than the introduction of antibiotics after the Second World War, further supports the importance of sexual behaviour in transmission dynamics. In future work, it would be interesting to integrate epidemiological evidence of sexual networks in purpose designed cohort studies to explore this further.

# Methods

## Samples

Overall ethical approval for receipt, handling and sequencing of all clinical samples, as well as for use of UK samples collected as part of public health surveillance and for research was granted by the London School of Hygiene and Tropical Medicine Observational Research Ethics Committee (REF#16014) and the National Health Service (UK) Health Research Authority and Health and Care Research Wales (UK; 19/HRA/0112). Samples were deidentified and not linked to any personal identifiable information. As no patient contact took place, no change to clinical care occurred and the study consists only in the use of residual DNA from samples which were already routinely collected, patient consent was deemed unnecessary during ethical approval. Ethical approval for sequencing the samples from Belgium was covered by a provision of the Institutional Review Board of the Institute of Tropical Medicine that allows the further characterization of residual patient samples without additional Ethical Committee clearance. In addition, at the Institute for Tropical Medicine outpatient clinic the patients are informed that their remnant samples may be used; if they do not consent they have a form to complete (opt out). Samples from Hungary were collected and preserved as part of the routine diagnostics (standard care), and stored at laboratories which have approval for preservation of such and other clinical samples, and no patient identification information was available - accordingly, these samples do not need a separate ethical approval for use in an anonymised manner. Samples from Russia were collected as part of a previous study that involved molecular epidemiology, and this had ethical approval from The State Research Center of Dermatology, Venereology and Cosmetology of The Russian Ministry of Health (SRCDVC), Moscow, Russia. Samples from South Africa were collected as part of a study on the impact of episodic acyclovir therapy on ulcer duration & HIV shedding from genital ulcers in men, and ethical approval was granted by the Human Research Ethics Committee of the University of the Witwatersrand in South Africa (Clearance Certificate Nos: M040548 and M10201). All

participants gave permission to store samples for future testing for infectious diseases. Zimbabwe samples were collected as part of the Zimbabwe STI Etiology Study, which had a provision for specimen storage and future studies and the consent form had a specific opt-in/opt-out addendum for specimen storage and future studies. It also specifically asked for consent to have specimens shipped to NICD in South Africa. The protocol and consent forms were approved by the Research Council and Medical Research Council of Zimbabwe. Samples from Canada (British Columbia and Alberta) were collected as part of public health surveillance, were deidentified before transfer between labs, and were deemed exempt from requiring additional ethical approval. Samples from Australia were covered by HREC approval, and that this approval included a waiver to obtain individual informed consent that was consistent with the requirements outlined in the Australian NHMRC National Statement. For samples from Spain, all the patients enrolled in provided written consent for collection of an additional ulcer swab and/or whole blood specimen to perform the TPA molecular studies. Institutional Review Board approval PR(AG)297/2014 was obtained from the Ethics Committee of Vall d'Hebron Research Institute. An amendment was also approved to allow WGS. For samples from Ireland, the study was approved by the ethics board of St James's Hospital and Tallaght Hospital, and this included approval for molecular analyses.

Novel samples from Australia (Brisbane, Melbourne), Belgium (Antwerp), Canada (Alberta, British Columbia), Hungary (National collection), Ireland (Dublin), Russia (Tuva Republic), South Africa (Johannesburg), Spain (Barcelona), Sweden (National collection), the UK (National collection), Zimbabwe (3 regions), were sequenced directly from genomic DNA extracted from clinical patient swabs or biopsies, typically utilising de-identified residual diagnostic samples which were further pseudonymised before analysis. Additional novel samples from Australia (Melbourne), Haiti and the USA (6 cities) were sequenced from historic freezer archives after prior passage in the rabbit model[4].

DNA extracts were quantified by qPCR (TPANIC_0574) as previously[4], and grouped into pools of either 32 or 48 with similar (within 2 $C_T$) treponemal load with high concentration outlier samples diluted as necessary. We added 4μl pooled commercial human gDNA (Promega) to all samples to ensure total gDNA > 1μg/35μl, sufficient for library prep.

## Sequencing

Extracted genomic DNA was sheared to 100-400 bp (mean distribution 150 bp) using an LE220 ultrasonicator (Covaris Inc). Libraries were prepared (NEBNext Ultra II DNA Library prep Kit, New England Biolabs, Massachusetts, USA) using initial adaptor ligation and barcoding with unique dual indexed barcodes (Integrated DNA Technologies, Iowa, USA). Dual indexed samples were amplified (6 cycles of PCR, KAPA HiFi kit, Roche, Basel, Switzerland), quantified (Accuclear dsDNA Quantitation Solution, Biotium, California, USA), then pooled in preassigned groups of 48 or 32 to generate equimolar pools based on Total DNA concentration. 500 ng pooled DNA was hybridised using 120-mer RNA baits (SureSelect Target enrichment system, Agilent technologies; Bait design ELID ID 0616571)[4,37]. Enriched libraries were sequenced on Illumina HiSeq 4000 to generate 150 bp paired end reads at the Wellcome Sanger Institute (Cambridgeshire, UK)

as previously described[38]. For one rabbit passaged sample from Melbourne, Australia (TPA_AUSMELT-1)[30], genomic DNA extracted from historically archived tissue lysate was sequenced on Illumina NextSeq 500 (150 bp paired end reads, Nextera DNA Flex libraries) without any prior enrichment to an estimated 1Gb/sample at the Doherty Institute (Melbourne, Australia).

## Sequence analysis and Phylogenetics

We filtered *Treponema* genus-specific sequencing reads using the full bacterial and human Kraken 2[39] v2.0.8 database (March 2019), followed by trimming with Trimmomatic[40] v0.39 and downsampling to a maximum of 2,500,000 using seqtk v1.0 (available at https://github.com/lh3/seqtk) as previously described[4]. For publicly available genomes, raw sequencing reads were downloaded from the European Nucleotide Archive (ENA) and subjected to the same binning and downsampling procedure. For five public genomes (see Supplementary Data 1), raw sequencing reads were not available; for these we simulated 150 bp PE perfect reads at 50X coverage from the RefSeq closed genomes using Fastaq v3.17.0 (available at https://github.com/sanger-pathogens/Fastaq).

For phylogenomic analysis, we mapped *Treponema*-specific reads to a custom version of the SS14_v2 reference genome (NC_021508.1), after first masking 12 repetitive Tpr genes (Tpr A-L), two highly repetitive genes (arp, TPANIC_0470), and five FadL homologs (TPANIC_0548, TPANIC_0856, TPANIC_0858, TPANIC_0859, TPANIC_0865) using bedtools[41] v2.29 maskfasta (positions listed in Supplementary Data 2). We mapped prefiltered sequencing reads to the reference using BWA mem[42] v0.7.17 (MapQ 20, excluding reads with secondary mappings) followed by indel realignment using GATK v3.7 IndelRealigner, deduplication with Picard MarkDuplicates v1.126 (available at http://broadinstitute.github.io/picard/), and variant calling and consensus pseudosequence generation using samtools[43] v1.6 and bcftools v1.6, requiring a minimum of two supporting reads per strand and five in total to call a variant, and a variant frequency/mapping quality cut-off of 0.8. Sites not meeting our filtering criteria were masked to 'N' in the final pseudosequence. After mapping and pseudosequence generation, we repeated the masking of the 19 genes on the final multiple sequence alignment using remove_block_from_aln.py available at https://github.com/sanger-pathogens/remove_blocks_from_aln/ to ensure sites originally masked in the reference were not inadvertently called with SNPs due to mapped reads overlapping the masked region. These 19 regions of recombination and genomic uncertainty due to gene orthology or repetitive regions[3,4,6] accounted for 30,071 genomic sites (Supplementary Data 2).

For basic lineage assignment of genomes, we excluded sequences with >75% genomic sites masked to 'N'. A SNP-only alignment was generated using snp-sites[44] v 2.5.1, and a maximum likelihood phylogeny was calculated on the variable sites using IQ-Tree[28] v1.6.10, inputting missing constant sites using the `-fconst` argument, and using a general time reversible (GTR) substitution model with a FreeRate model of heterogeneity[45] and 1000 UltraFast Bootstraps[46].

For finescale analysis of high-quality genomes, we excluded sequences with >25% genomic sites masked to 'N' (i.e. >75% genomic sites passing filters at >5x and not masked). We

*Nat Microbiol*. Author manuscript; available in PMC 2021 December 03.

used Gubbins[27] v2.4.1 (20 iterations) to generate recombination-masked full genome length and SNP-only alignments. Gubbins[27] identified 19 further putative regions of recombination affecting 2.1% of genomic sites (n=23,567) and 27 genes (listed in Supplementary Data 2), meaning we masked a maximum of 4.7% (53,638 sites) of the genome over the 38 regions. We used IQ-Tree on the SNP-only alignment containing 901 variable sites, inputting missing constant sites using the `-fconst` argument, and allowing the built-in model test to infer a K3Pu+F+I model and 10,000 UltraFast bootstraps.

To cluster genomes, we initially performed joint ancestral reconstruction[47] of SNPs on the phylogeny using pyjar v0.1.0 (available at https://github.com/simonrharris/pyjar), and used this to determine phylogenetic clusters with a 10 SNP threshold in rPinecone[29] v0.1.0 (available at https://github.com/alexwailan/rpinecone). We further investigated this by using IQ-tree to generate 100 standard non-parametric bootstraps on the maximum likelihood phylogeny, and used the resulting 100 trees as independent inputs to rPinecone, as described in the rPinecone manuscript[29]. We used the hierarchical clustering 'hclust' algorithm in R[48] to group rPinecone clusters, and evaluated different proportions of trees supporting clusters against the phylogeny (Supplementary Figure 2).

To assess the impact of missing sites in the multiple sequence alignments used to construct our phylogenies, we subsampled the recombination-masked multiple sequence alignment to only include 301 genomes with <5% of genomic positions masked to 'N', and repeated the maximum likelihood analysis using IQ-Tree. We converted the resulting phylogeny to an ultrametric tree using phytools[49] v0.7-47; comparison with the ultrametric tree of our finescale analysis of 528 genomes in a tanglegram indicated that the underlying topology and sublineage clusters were the same (Supplementary Figure 7), apart from the low diversity sublineage 1 region. To assess the impact of mapping to an alternative reference genome, we also mapped the 528 genomes to the Nichols_v2 reference (CP004010.2), repeating the recombination masking and phylogenetic reconstruction as described above. Comparison of the resulting Nichols-mapped ultrametric tree to our SS14-mapped tree in a tanglegram also indicated equivalent topology and clustering of sublineages (Supplementary Figure 8), with the exceptions of sublineage 1 (which has low diversity, and thus low support for within-group topology), and sublineage 6 (which diverges from other strains close to the root of TPA, and appears on either the Nichols- or SS14-Lineage side of different midpoint rooted trees).

For temporal analysis, our dataset was too large for robust model testing of all genomes, so we stratified our dataset by sublineage and country, then used the random sampler in R[48] to select a maximum of five genomes from each strata, plus all singleton strains, yielding a dataset of 138. We extracted the sequences from the multiple sequence alignment using seqtk and the subtree from our broader phylogeny using ape[50] v5.4.1 `keep.tip`. Root-to-tip distance analysis from this subtree showed a correlation of 0.327 and $R^2$ of 0.11 (Supplementary Figure 9), and we proceeded to BEAST analysis. We initially ran BEAST[31,51] v1.8.4 on our recombination-masked SNP-only alignment containing 592 variable sites, correcting for invariant sites using the constantPatterns argument, in triplicate using both a Strict Clock model[52] (starting rate prior 1.78 x$10^{-7}$) and an Uncorrelated Relaxed Clock model[53], with HKY substitution model[54] and diffuse gamma distribution

prior[55] (shape 0.001, scale 1000) over 100 million MCMC cycles with 10 million cycle burnin. We evaluated constant, relaxed lognormal, exponential and Bayesian Skyline (10 categories) population distributions[56]. All MCMC chains converged with high effective sample sizes, and on inspection of the marginal distribution of ucld.stdev we could not reject a Strick Clock. We used the marginal likelihood estimates from the triplicate BEAST runs as input to Path Sampling and Stepping Stone Sampling analysis[57,58], and this suggested that the Strict Clock with Bayesian Skyline was the optimal model for this dataset (Supplementary Figure 10), with an inferred molecular clock rate of $1.23 \times 10^{-7}$ substitutions/site/year. This is consistent with previous molecular clock rate estimates for TPA[3,4], but we note that inclusion of recombinogenic or hypervariable sites outside the clonal frame (masked in this study) would be expected to increase this rate. To ensure that our findings were not artefactual to the down-sampled dataset, we re-stratified the dataset by sublineage, country and year, selecting a maximum of 3 genomes from each stratum, plus all singleton strains, yielding a dataset of 168 genomes with 466 variable sites. We ran BEAST on this new subsampled dataset using the optimal Strict Clock (with starting rate prior of $1.23 \times 10^{-7}$, inferred from the previous analysis) with Bayesian Skyline from above over 100 million MCMC cycles, with the equivalent results (Supplementary Figure 11).

To evaluate the temporal dynamics of sublineages, we tested the temporal signal for the 4 largest sublineages 1, 2, 8, and 14 (Supplementary Figure 4). Sublineage 14 had poor temporal signal and was excluded from further analysis. We performed independent BEAST analyses on the remaining sublineage multiple sequence alignments using the optimal Strict Clock model with Bayesian Skyline (10 population groups) described above, evaluating 3 independent chains of 200 million cycles for each.

To analyse the full dataset (n=520 after excluding heavily passaged samples or those with uncertain collection dates, 883 variable sites), after evaluating temporal signal (Supplementary Figure 12), we initially attempted to reproduce our model in BEAST 1.8.4, but this proved unachievable with our local implementation and compute arrangements. To analyse the full dataset, we therefore reconstructed the optimal BEAST v1.8.4 model (Strict Clock with reference rate prior of $1.23 \times 10^{-7}$s/s/y, HKY substitution model[54], Coalescent Bayesian Skyline distribution with 10 populations[54,56]) in a BEAST2[59] v2.6.3 implementation with BEAGLE[51] libraries optimised for Graphical Processing Units, analysing the 520 genomes over 500 million MCMC cycles in triplicate. To compare the phylodynamics of the individual Nichols- and SS14-lineages, we repeated the BEAST2 analysis described above using multiple sequence alignments specific to Nichols (n=94) and SS14 (n=426); we used Tracer[31] v1.7.1 to extract Bayesian Skyline distributions and lineage accumulation for plotting in R.

To further confirm the temporal signal in our full 520 genome tree, we used the TIPDATINGBEAST[60] v1.1-0 package in R to perform a date randomisation test, generating 20 new datasets with randomly reassigned dates from the original xml file and conditions – BEAST2 analysis using the same prior conditions found no evidence of temporal signal in these replicates, indicating that the signal in our tree was not found by chance (Supplementary Figure 13).

We used logcombiner v2.6.3 with a 10% burnin to generate consensus log and treefiles, resampling 100,000 states for the full 520 sample analysis, and treeannotator v1.8.4 to create median maximum credibility trees. We generated Bayesian Skyline and Lineage accumulation plots using the combined log and tree files in Tracer v1.7.1[31], exporting the data for subsequent plotting in R. To evaluate the posterior distribution of population expansion times, we used the script population_increase_distribution_BEAST.py (available at https://github.com/chrisruis/tree_scripts commit:2463656e329e3f25ec6dd13c86c64ad163525ae0), which uses the BEAST log and tree files to identify the first increase in relative genetic diversity from the PopSizes columns and the date of this increase using the corresponding number of nodes in the GroupSizes columns and the node heights in the respective tree. We required a 2-fold population expansion (defined by setting `-p` to 100). The script outputs the proportion of trees in the posterior distribution that support an increase in relative genetic diversity, along with the distribution of expansion dates, which we plotted in R[48]. We repeated this analysis using the script population_change_support_BEAST.py (available at https://github.com/chrisruis/tree_scripts), which looks for an increase or decrease of effective population size within a defined window, testing for supported start dates of a 2-fold population decline or expansion between 1990 and 2015.

For analysis of genetic changes between common ancestral nodes in our phylogeny, we performed ancestral reconstruction of the full 528-sample Maximum Likelihood alignment and tree using TreeTime[61] v0.7.4. We extracted SNPs from the resulting multiple sequence alignment using snp-sites, functionally annotated variants using SnpEff[62] v4.3 with the most recent NCBI annotation for the NC_021508.1 SS14 reference genome (June 2021), and imported data into R[48] v3.6.0 for analysis using vcfR[63] v1.12.0. We selected the common ancestral nodes leading to contemporary SS14- and Nichols-lineages (see Supplementary Figure 6), and used R to extract annotated variants that differed between the relevant nodes from our VCF.

For comparison of TPA sublineage trends with national syphilis rates, we downloaded and plotted publicly available incidence data for England (https://www.gov.uk/government/statistics/sexually-transmitted-infections-stis-annual-data-tables) and British Columbia (http://www.bccdc.ca/health-professionals/data-reports/sti-reports).

Macrolide resistance alleles were inferred using the competitive mapping approach previously described[4,38] (available at https://github.com/matbeale/Lihir_Treponema_2020/competitive_mapping_Treponema23S--mod.sh commit:044b29ce29ada81e4f7cb0318301e97e1d5a8d55). To infer pairwise SNP distances between samples, we used pairsnp v0.1.0 (available at https://github.com/gtonkinhill/pairsnp commit: 0acddba060cc076946dab9969a95ab3c21f110fb). We constructed networks of minimum pairwise distance and shared lineages in R, and plotted these as heatmaps using ggplot2[64] v3.3.2. Nucleotide diversity ($\pi$) for different clades was inferred from the multiple sequence alignments using EggLib[65] v3.0.0b21, including variable sites present in at least 5% of genomes. For geospatial analysis, we used the CoordinateCleaner[66] v2.0-17 package in R to define the centroid position for each country, apart for Russia (where we used the centroid of the Tuva Republic) and Mexico (where we used Mexico City). Geographic

distances between countries (using the country centroid or location defined above) were determined using the distVincentyEllipsoid function from the geosphere[67] v1.5-10 package. Correlations between pairwise genetic, geographic and temporal distance were inferred using two-sided Pearson's R Correlation via the 'cor' function in R, where we compared 'real' correlation with 1000 bootstraps resampled with replacement to obtain a p value. Sample counts were plotted using ggmap[68] v3.0.0 over maptiles downloaded from Stamen Design (http://maps.stamen.com). All phylogenetic trees were plotted in R using ggtree[69] v2.5.1. All figures used ggplot2[64] for plotting, and multi-panel figures were constructed using cowplot[70] v1.1.1.

## Extended Data

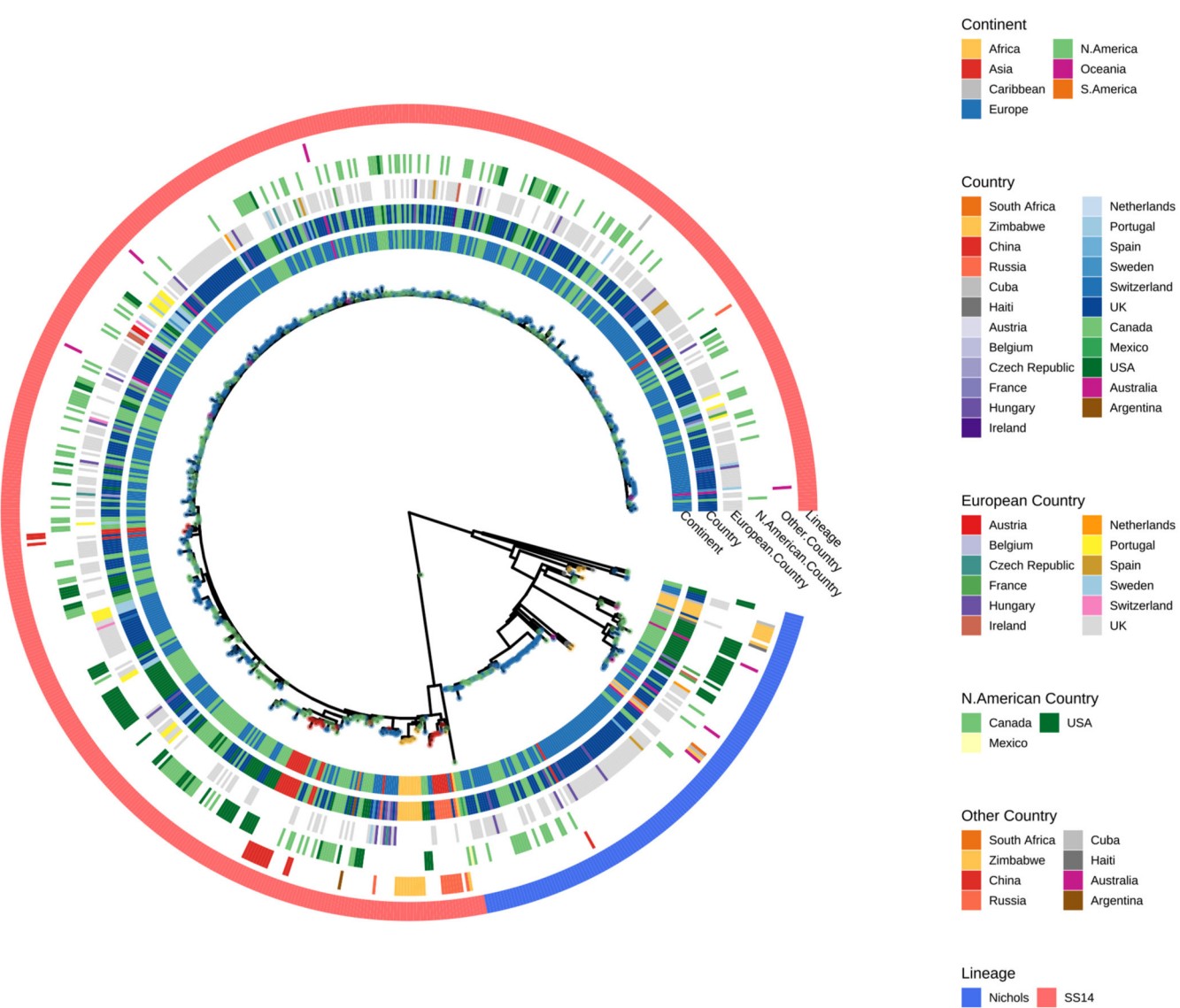

**Extended Data Fig. 1. Phylogenetic distribution of 726 *Treponema pallidum* ssp *pallidum* partial genomes.**

Maximum likelihood phylogeny of 726 partial (>25% of genome positions) genomes shows two primary lineages (Nichols, SS14), with no obvious correlation by country or continent. Tree tip points are coloured by continent. Coloured strips show continent, country (all), countries separated by region (European Countries, North American Countries, Other Countries) and TPA lineage. One very low coverage sample (TPA_BCC144, Canada, 47% genome breadth, 7.9X mean coverage) appears basal to the SS14-lineage clade in this phylogeny, but due to low coverage it was not possible to determine the correct placement.

## SS14-lineage phylogeny

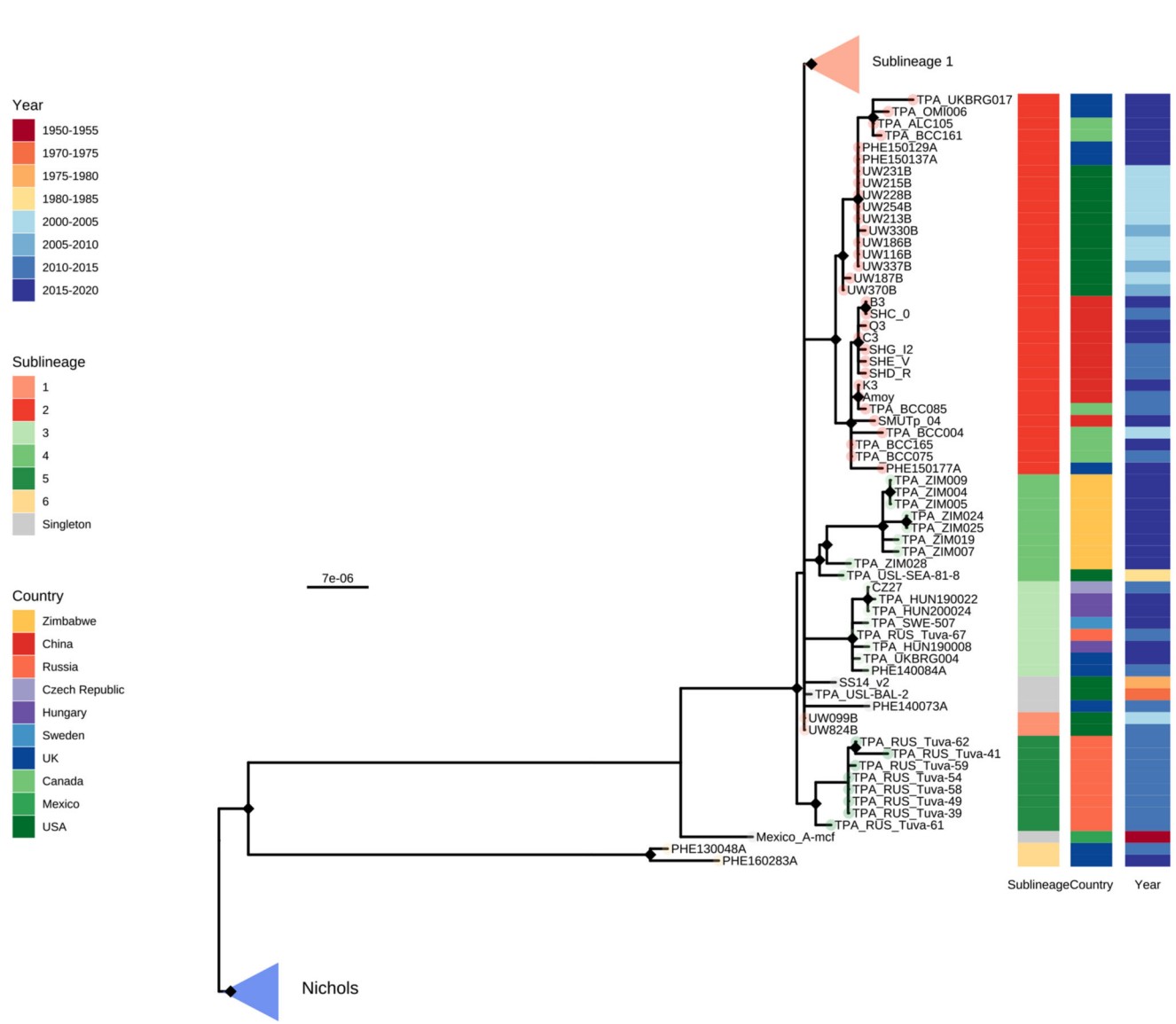

**Extended Data Fig. 2. Detailed subtree of SS14-lineage.**

Recombination masked WGS phylogeny, showing the SS14-lineage and sublineages. The low diversity globally distributed sublineage 1 has been collapsed to enable visualization of the remaining sublineages. Tip points are coloured by sublineage, and coloured strips show sublineage and country. Blue triangle indicates collapsed Nichols-lineage, pink triangle indicates collapsed sublineage 1. Two samples close to the root of the common SS14-lineage clades were clustered as sublineage 1, and are shown. Note that sublineage 6 diverges from a node close to the root of TPA, and appears on the SS14 side in this midpoint rooted tree.

Nichols-lineage phylogeny with collapsed nodes

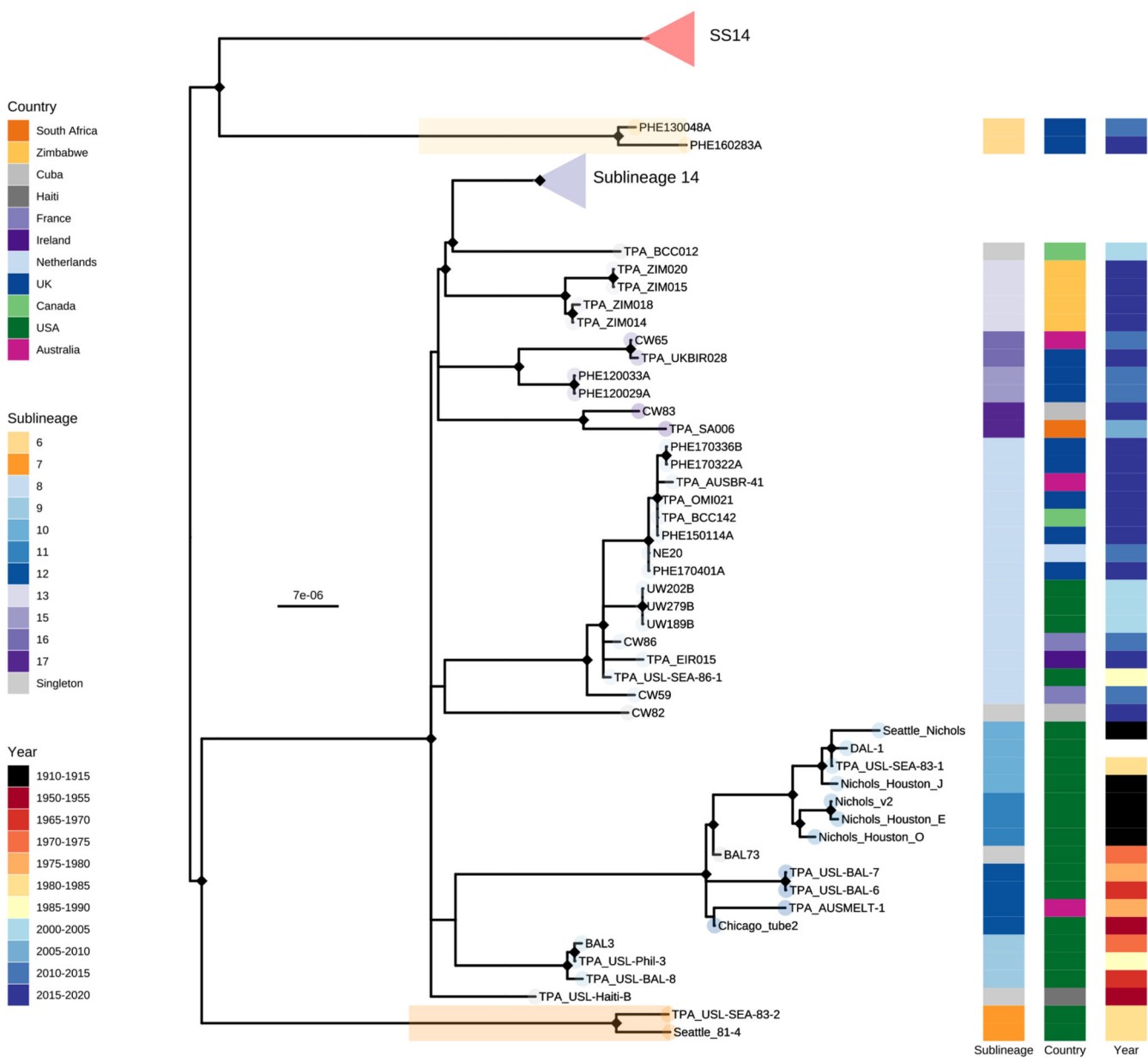

**Extended Data Fig. 3. Subtree highlighting novel Nichols-lineage strains.**

*Nat Microbiol.* Author manuscript; available in PMC 2021 December 03.

Recombination masked WGS phylogeny, showing the Nichols-lineage and sublineages. Tip points are coloured by sublineage, and coloured strips show sublineage and country. Shaded boxes highlight basal Nichols-lineage outgroup sublineages 6 and 7. Note that sublineage 6 diverges from a node close to the root of TPA, and appears on the SS14 side in this midpoint rooted tree. The large clonal sublineage 14 has been collapsed to enable clearer visualization of the remaining taxa. The pink triangle indicates collapsed SS14-lineage, blue triangle indicates the collapsed sublineage 14.

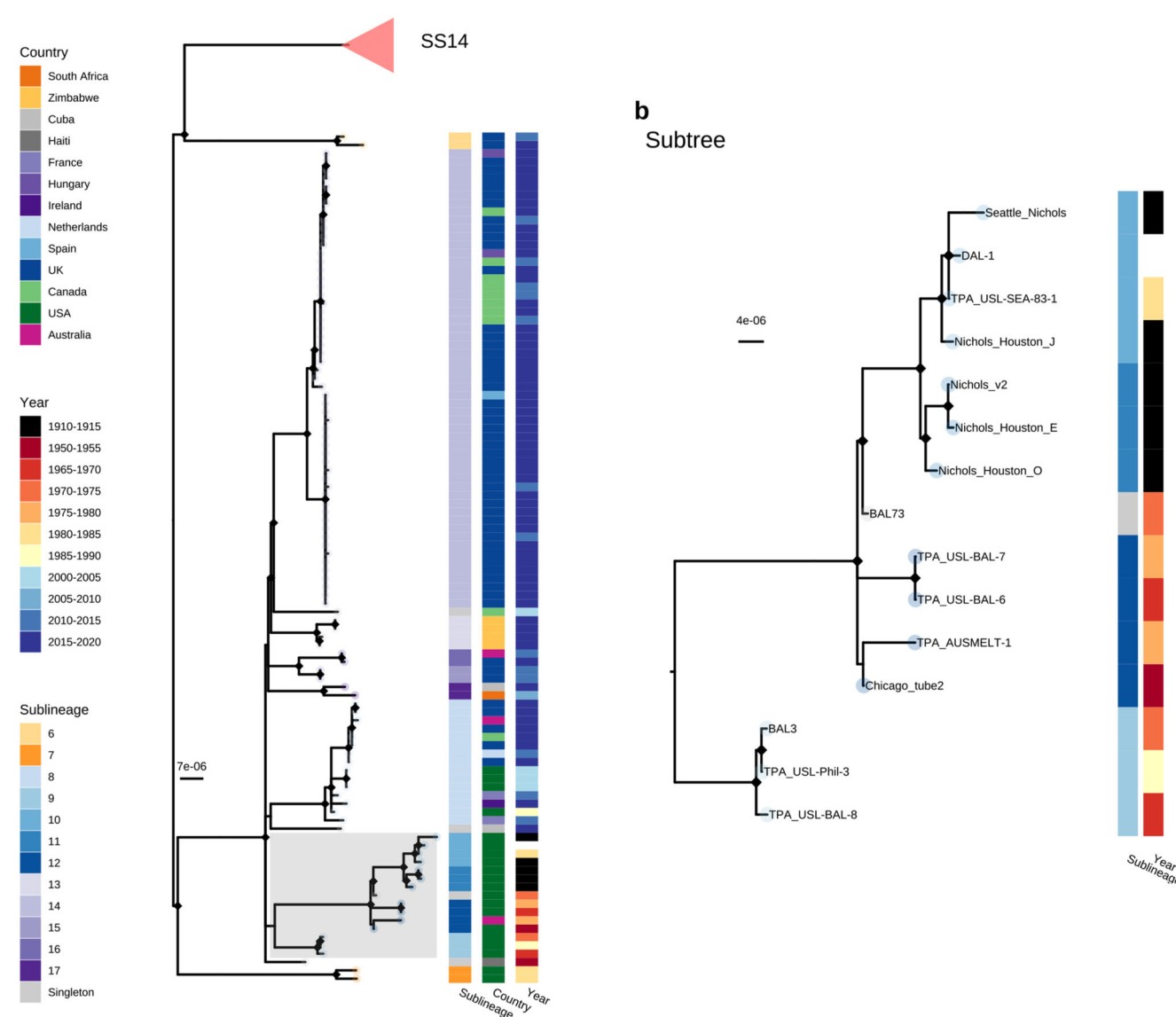

**Extended Data Fig. 4. Commonly used Nichols Reference genomes form a monophyletic clade distinct from contemporary clinical strains.**

A-Recombination masked WGS phylogeny, showing the Nichols-lineage and sublineages. Shaded grey box shows a monophyletic clade containing commonly used reference genomes as well as genetically related strains. Tip points are coloured by sublineage, and coloured strips show sublineage and country. Pink triangle indicates collapsed SS14-lineage. B-Expanded view of a seemingly extinct clade containing common reference strains including Nichols_v2, DAL-1 and Seattle_Nichols. The most recent sample closely related to the reference strains (TPA_USL-SEA-83-1) was collected in 1983, while the latest sample for the entire clade (TPA_USL-Phil-3) was collected in 1987. The provenance of the sample originally used for sequencing the DAL-1 genome is uncertain, but in the literature the original isolation was in 1988. The placement of both DAL-1 and TPA_USL-SEA-83-1 within the diversity of Nichols-1912 derivatives suggests the possibility of the samples being mislabeled in the handling laboratories.

**a**

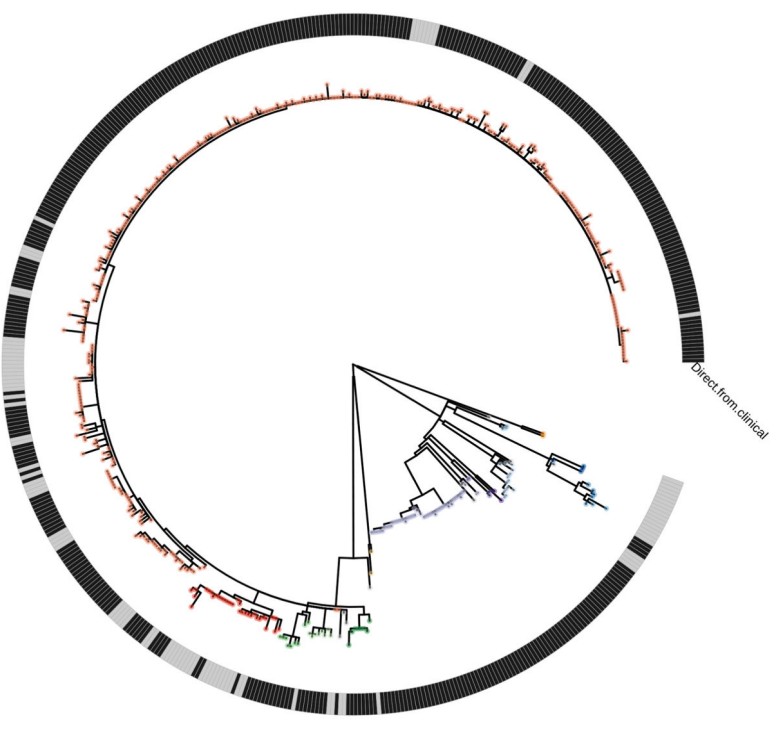

**b**

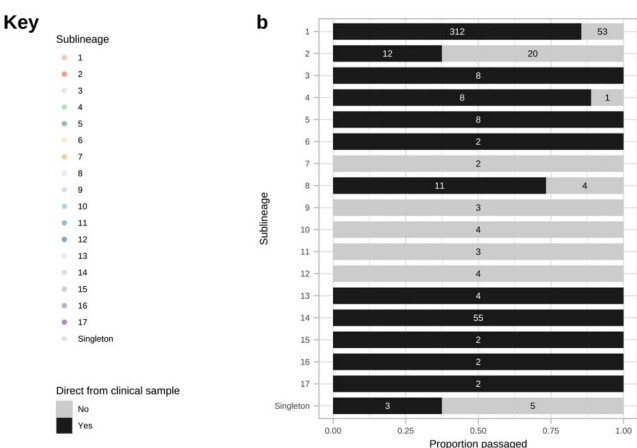

**Extended Data Fig. 5. Finescale analysis of 528 high quality TPA genomes and sublineages, showing distribution of samples sequenced directly from clinical samples and those passaged in rabbit model.**

A – Whole genome phylogeny showing distribution of samples sequenced directly from clinical sample or rabbit-passaged. B – Distribution of samples sequenced directly from clinical sample and rabbit-passaged samples according to sublineage, showing proportion (bar) and exact count (number). For both A and B, plots are coloured according to being directly sequenced from clinical samples (Black) or after rabbit passage (grey). Samples passaged in rabbits are distributed throughout the global TPA phylogeny, and present in

9/17 sublineages. Older samples from before 2000 were isolated via rabbit passage, and dominate extinct clusters, as well as clustering close to the most recent common ancestor of contemporary sublineages such as SS14 sublineage 1.

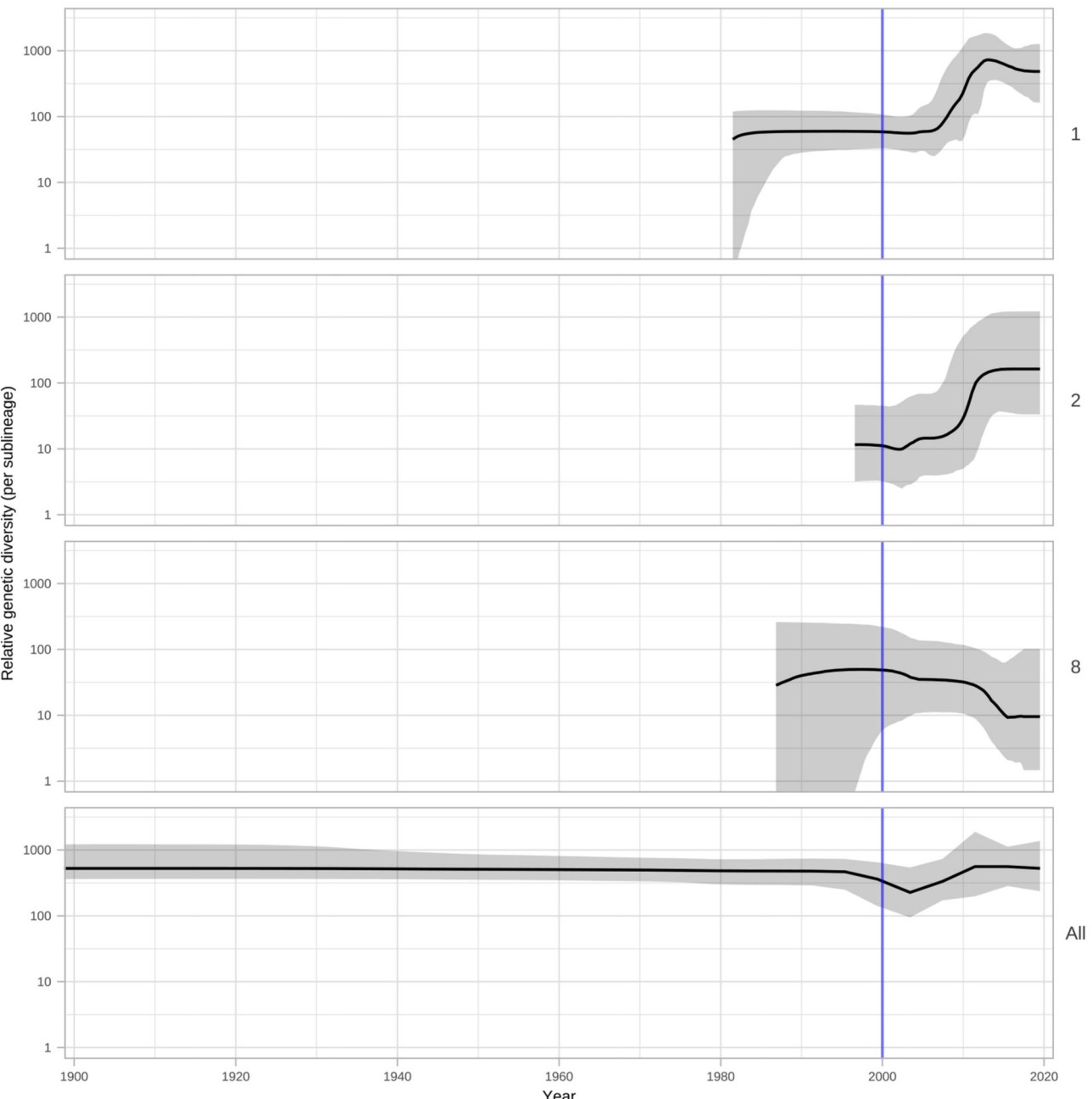

**Extended Data Fig. 6. Bayesian Skyline analysis of sublineages.**
Plots show population expansions occurring during the early 2000s for all sublineages with >15 samples apart from sublineage 14. Sublineage 14, which had low temporal signal, did

*Nat Microbiol.* Author manuscript; available in PMC 2021 December 03.

not converge after multiple BEAST runs. Shows Skyline plots of sublineages 1, 2, 8 and plot for all samples from Figure 3.

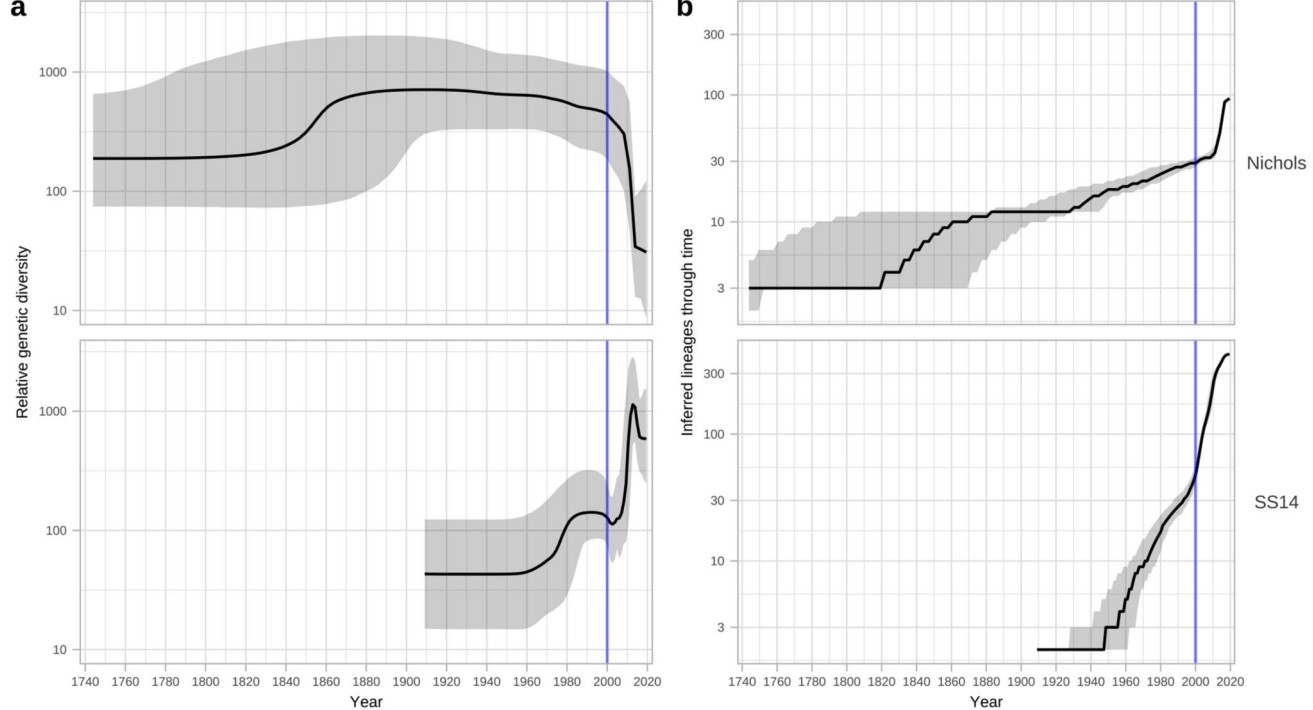

**Extended Data Fig. 7. Independent analysis of SS14 and Nichols phylodynamics shows differential patterns of expansion and decline.**

A – Independent Bayesian Skyline plots for Nichols- and SS14- lineages. B – Independent lineages through time plot for Nichols- and SS14-lineages. Skyline analysis indicates expansion of SS14-lineage after 2000 coincided with a decline of Nichols-lineage. However, analysis of lineage accumulation through time shows that both SS14 and Nichols continued to expand after 2000; whilst this is visible as a steep slope from 2000 in SS14, rapid expansion for Nichols occurred after 2010.

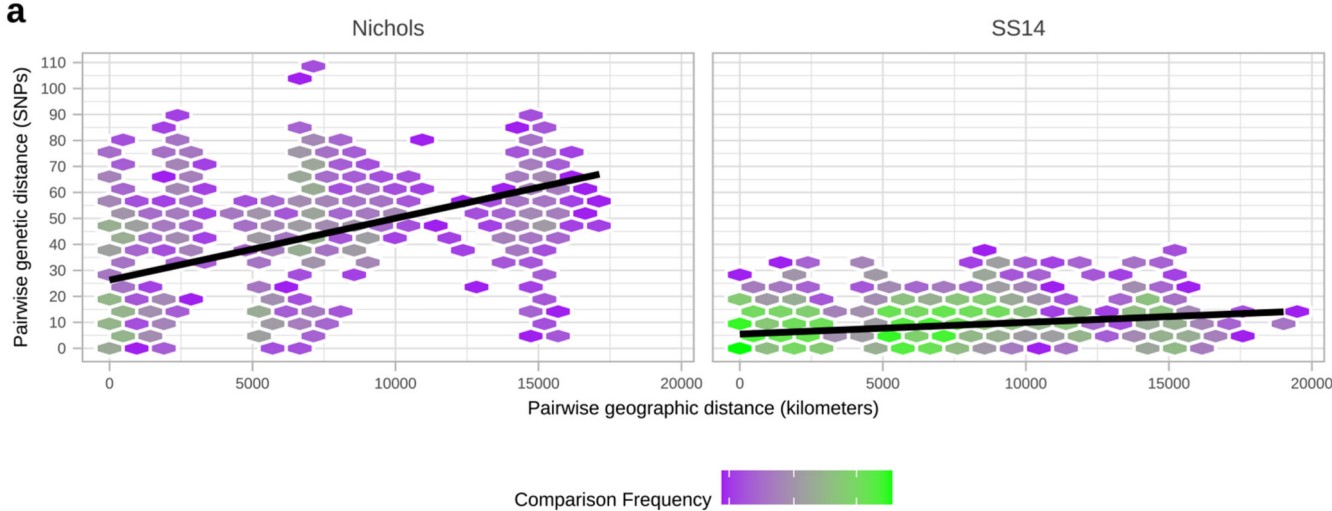

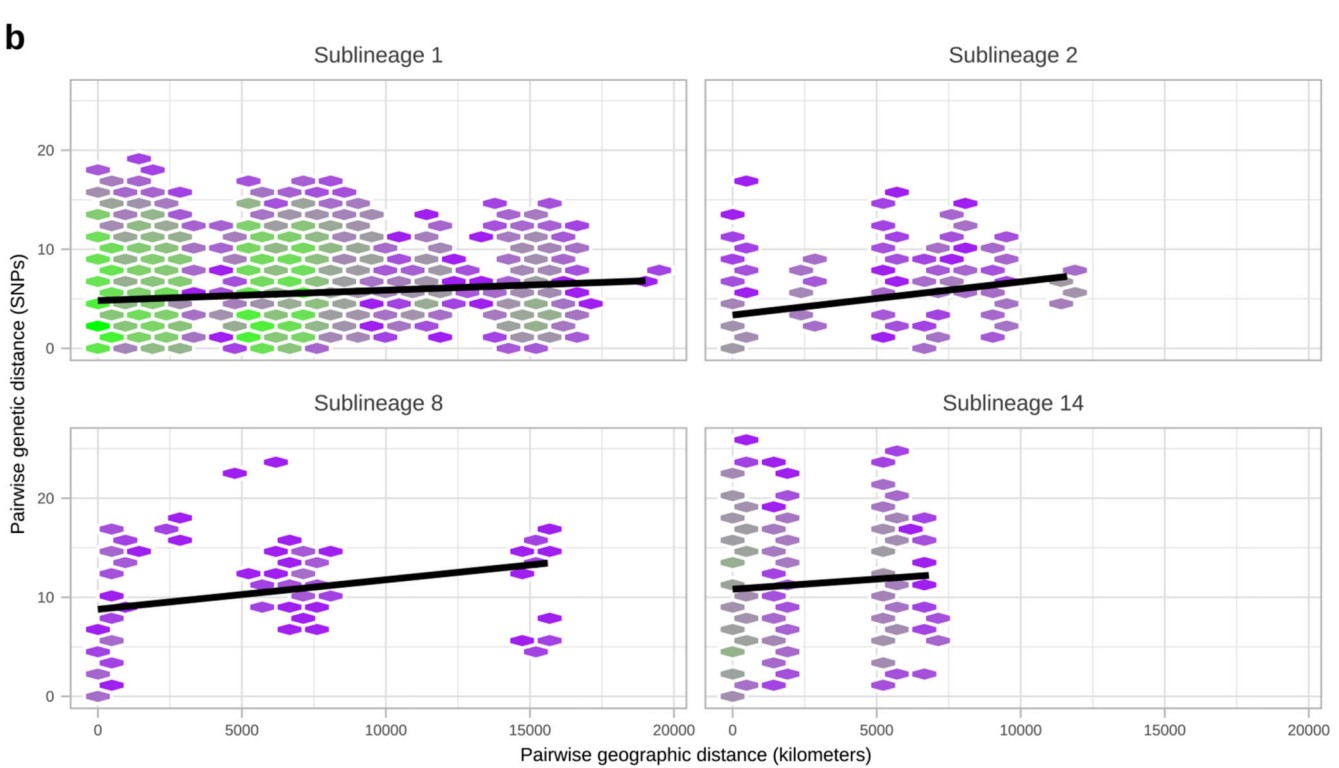

**Extended Data Fig. 8. Effect of geographic distance on genetic distance.**
A-Pairwise comparison of genetic distance (SNPs) and geographic distance (km; calculated using country centroids) within Nichols- and SS14-lineages, including linear regression (95% CI not visible). B- Pairwise comparison of genetic distance (SNPs) and geographic distance (km; calculated using country centroids) within the four major multi-country sublineages (SS14: 1, 2; Nichols: 8, 14). Includes linear regression (95% CI not visible).

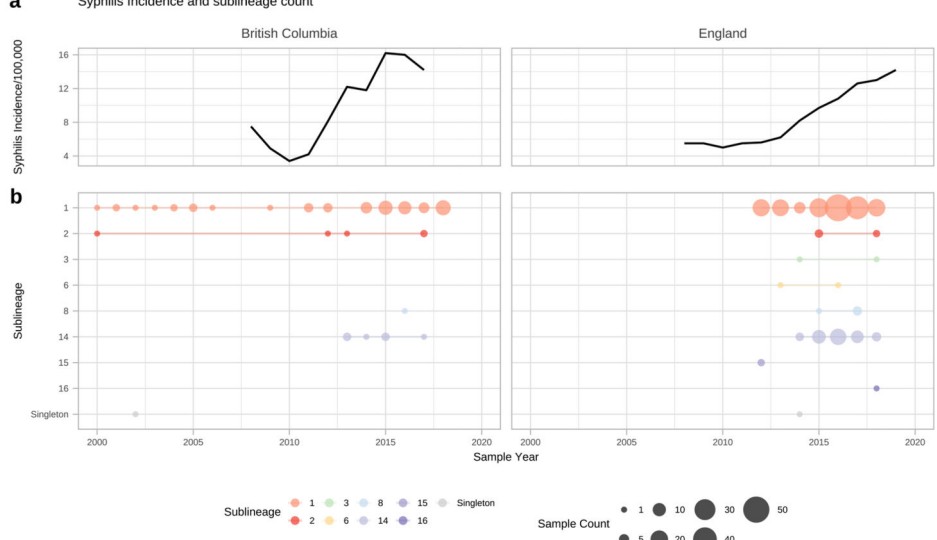

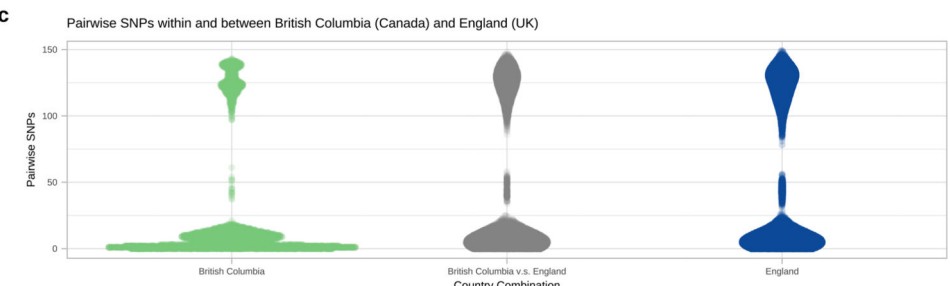

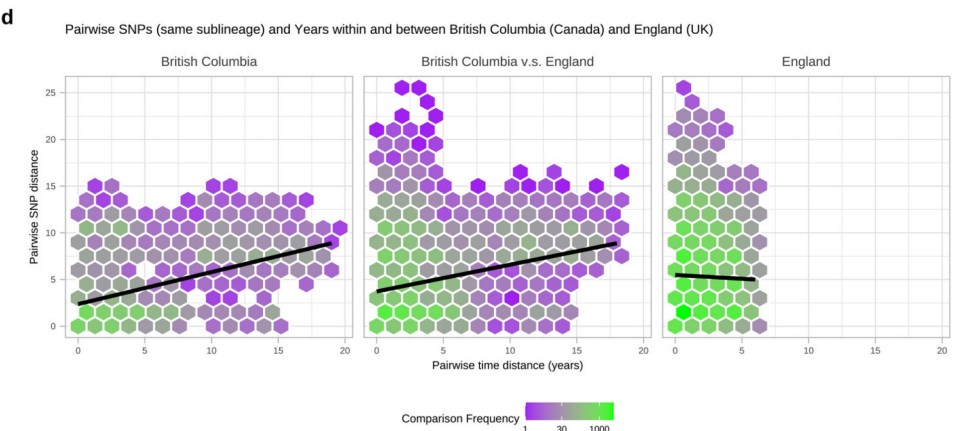

**Extended Data Fig. 9. Sharing of sublineages and closely related strains within and between British Columbia (Canada) and England (UK).**

A-Syphilis incidence per 100,000 population by year for British Columbia, (Canada) and England (UK) using currently published data. B- TPA sublineage counts for each year, using high quality genomes from British Columbia (n=84) and England (n=240). British Columbia samples collected from 2000-2018, English samples collected from 2012-2018. C- Pairwise comparison of SNP distance distributions from samples within and between British Columbia and England. D- Comparison of SNP distance and temporal distance within and between British Columbia and England. The plot is divided into hexagonal bins,

with the colour of each hexagon representing the number of comparisons (white=none, purple=few, green=many, see scale). Linear regression lines also shown (95% CI not visible).

**a**

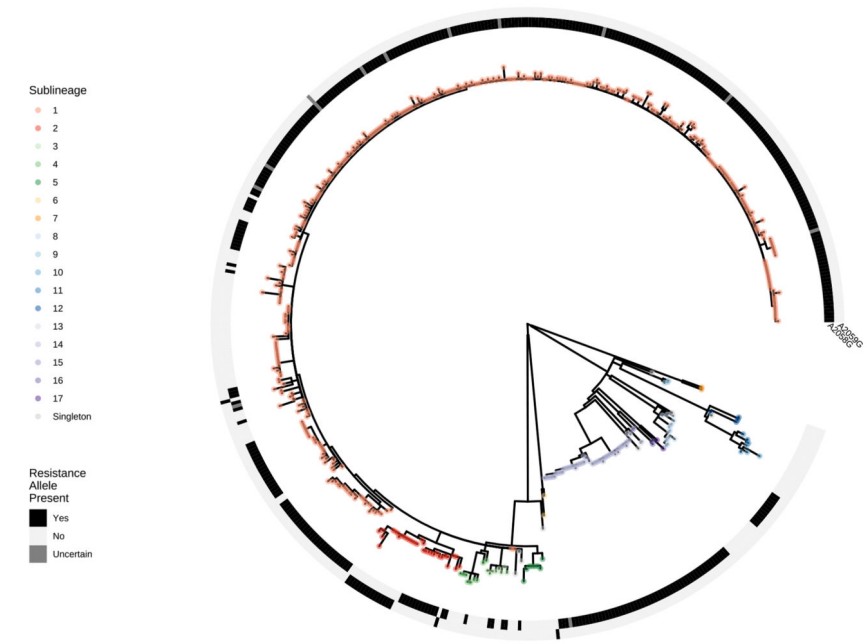

**b**

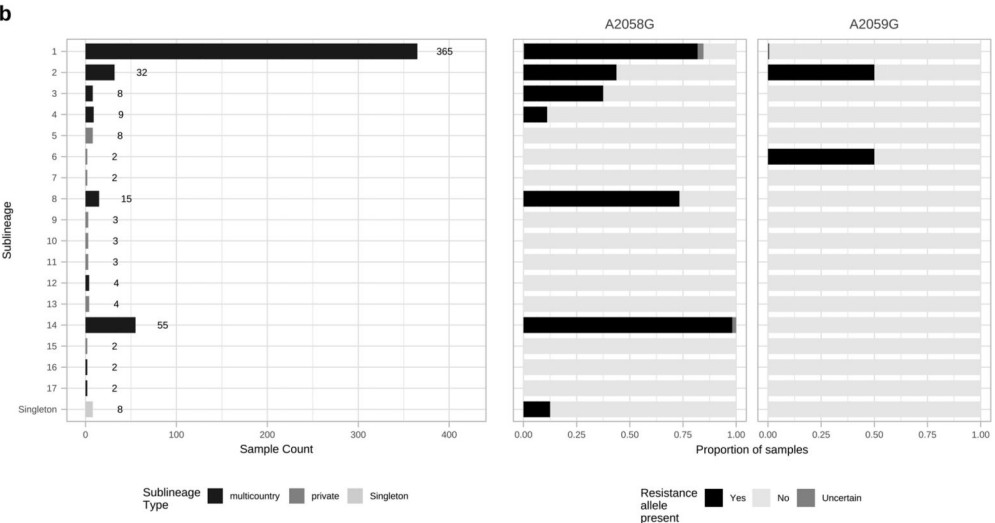

**Extended Data Fig. 10. Multicountry sublineages are broadly macrolide resistant.**
A-Whole genome phylogeny showing distribution of macrolide resistance conferring SNPs (A2058G and A2059G). B- Distribution of macrolide resistance SNPs by sublineage, indicating number of samples per sublineage, and sublineage type. Note that while the

common A2058G mutation was found in six sublineages (both Nichols- and SS14-lineages), we also found the less common A2059G in both SS14-lineage (sublineages 1, 2) and Nichols-lineage (sublineage 6).

## Supplementary Material

Refer to Web version on PubMed Central for supplementary material.

## Acknowledgements

The authors acknowledge the sequencing team at the Wellcome Sanger Institute, and Christoph Puethe and the Pathogen Informatics team for computational support. We thank additional technical staff involved in sample diagnostics, DNA extraction and sample retrieval in laboratories at Public Health England and NHS laboratories, UK; British Colombia CDC and Alberta Precision Laboratories, Canada; National Public Health Center, Budapest, Hungary; FRC Kazan Scientific Center, Tuva, Russia; National Institute for Communicable Diseases, Johannesburg, South Africa; Institute of Tropical Medicine, Antwerp, Belgium; Sahlgrenska University Hospital, Gothenburg, Sweden; Hospital Vall d'Hebron, Barcelona, Spain; Midlands Regional Hospital Portlaoise, Ireland; Pathology Queensland Central Laboratory, Australia; WHO Collaborating Centre for Gonorrhoea and other STIs, Sweden. We thank G. Tonkin-Hill, A. van Tonder, and members of the Thomson team for helpful discussions during analysis. MAB and NRT are supported by Wellcome funding to the Sanger Institute (#206194). MM is funded by the UKRI and NIHR [COV0335; MR/V027956/1, NIHR200125] and the EDCTP [RIA2018D-249]. DMW is funded by a Queensland Advancing Clinical Research Fellowship from the Queensland Government. DAW is supported by an Investigator Grant (1174555) from the National Health and Medical Research Council of Australia. SAL is funded by the National Institutes of Health (R01 AI42143 and R01 AI123196). This research was funded in whole, or in part, by the Wellcome Trust (#206194). For the purpose of Open Access, the author has applied a CC-BY public copyright licence to any Author Accepted Manuscript version arising from this submission.

## Data Availability

Sequencing reads for all novel genomes have been deposited at the European Nucleotide Archive (https://www.ebi.ac.uk/ena/browser/home) in BioProjects PRJEB28546, PRJEB33181 and PRJNA701499. All accessions, corresponding sample identifiers and related metadata are available in Supplementary Data 1. Map tiles were downloaded from http://maps.stamen.com using the ggmap interface. Publicly available syphilis incidence data is available for England at https://www.gov.uk/government/statistics/sexually-transmitted-infections-stis-annual-data-tables and for British Columbia at http://www.bccdc.ca/health-professionals/data-reports/sti-reports.

All sample metadata and intermediate analysis files are available at DOI:10.6084/m9.figshare.14376749 and https://github.com/matbeale/Contemporary_Syphilis_Lineages_2021. The minimum raw datafiles required to construct the Main and Extended Figure are described in Supplementary Data 4. The finescale maximum likelihood phylogeny and metadata are also available for interactive visualisation at https://microreact.org/project/xt7AuLJorkyBNHVXL2sF8G/1a515b2c.

## Code Availability

R code used for all statistical analysis and plotting is available in an Rnotebook at DOI:10.6084/m9.figshare.14376749 and at https://github.com/matbeale/Contemporary_Syphilis_Lineages_2021, along with underlying source files.

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

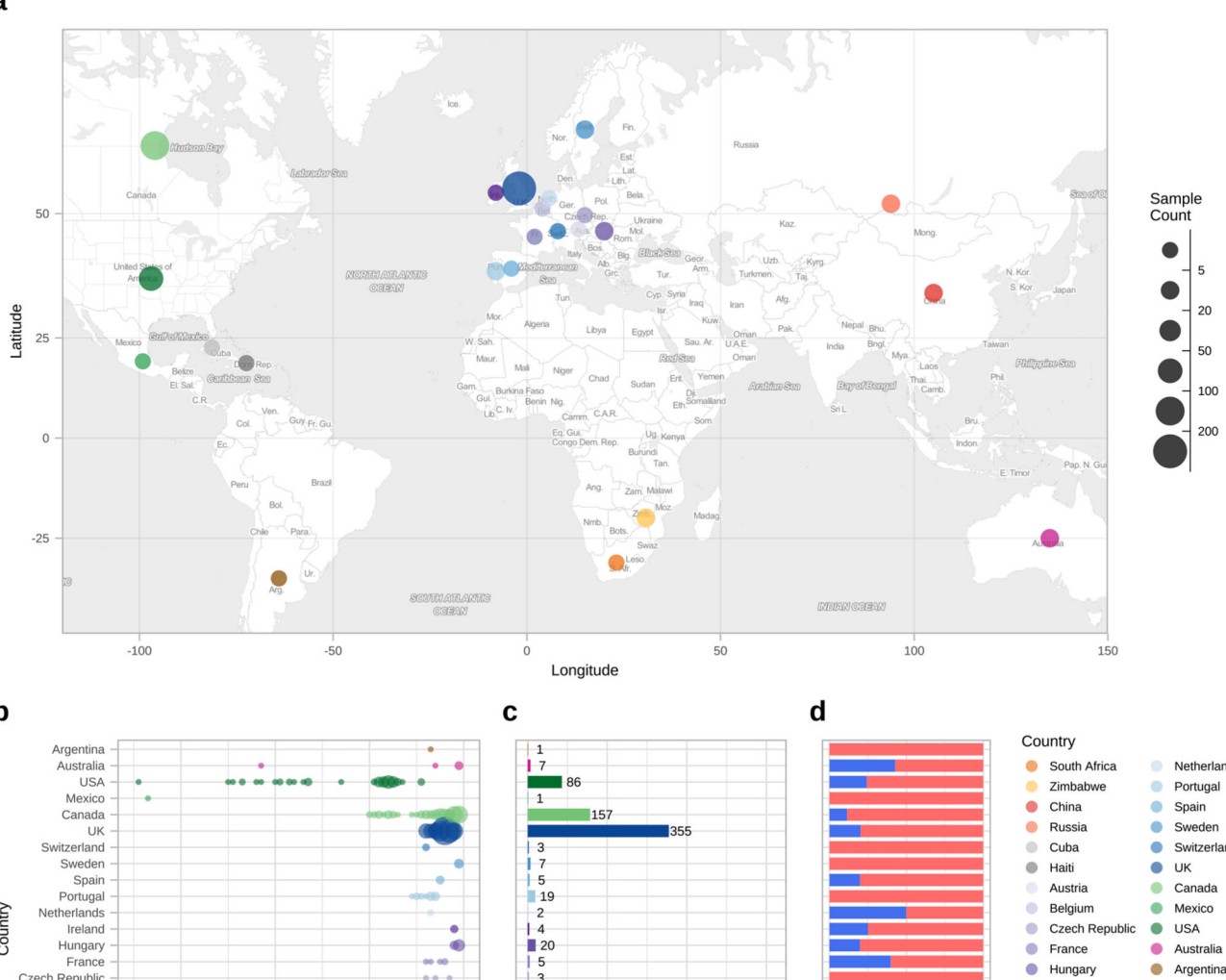

**Figure 1. Global distribution of 726 *Treponema pallidum* subspecies *pallidum* partial genomes.**
A- Map of sampled countries for 726 partial (>25% of genome positions) genomes.
Circle size corresponds to total number of genomes (binned into categories), and colour
corresponds to country. Country coordinates used are the country centroid position, apart
from Russia (where the centroid for the Tuva Republic is used) and Mexico (where the
location of Mexico City is used). Map tiles by Stamen Design (CC-BY 3.0), map data by
OpenStreetMap (ODbl). B- Temporal distribution of samples by country. Size of circle
indicates number of samples for that year. Three samples from Baltimore (USA) had

*Nat Microbiol.* Author manuscript; available in PMC 2021 December 03.

uncertain sampling dates (1960-1980) and were set to 1970 for plotting dates (1B, 1E). Genomes derived from passaged variants of the Nichols-1912 isolate or with uncertain collection dates are not shown in plotted timeline (1B, 1E). C- Total count of samples by country. D- Relative proportion of country samples corresponding to each TPA lineage (where only one sample was present per country, shows the lineage this corresponds to). E-Temporal distribution of the samples by TPA lineage. F- Total count of samples by TPA Lineage.

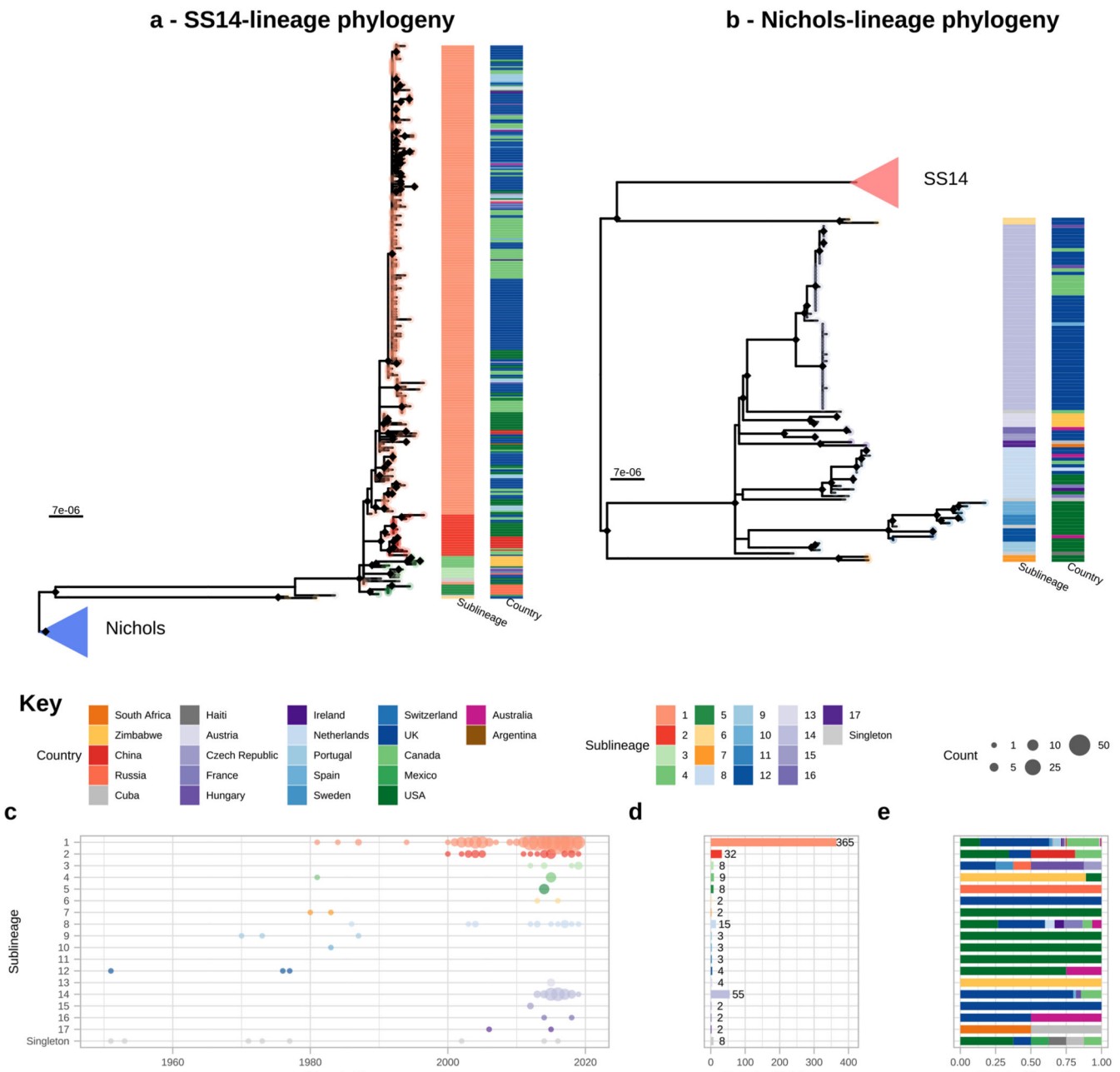

**Figure 2. Fine scaled analysis of 528 high quality (>75% reference sites) TPA genomes and sublineages.**

A- Recombination masked WGS phylogeny, highlighting the SS14-lineage (n=426).

B- Recombination masked WGS phylogeny, highlighting the Nichols-lineage (n=102), including four outlying genomes (sublineages 6 & 7). For A and B, coloured strips show sublineage and country; Tree tips show sublineage. Coloured triangle indicates node position of collapsed sister lineage. UF Bootstraps >=95% are marked with black node marks. Note that sublineage 6 is shown in both trees (see main text). C- Temporal distribution of samples by sublineage (unrelated singleton genomes are grouped together). D- Total count

of samples by sublineage. E- Relative proportion of each sublineage samples corresponding to country.

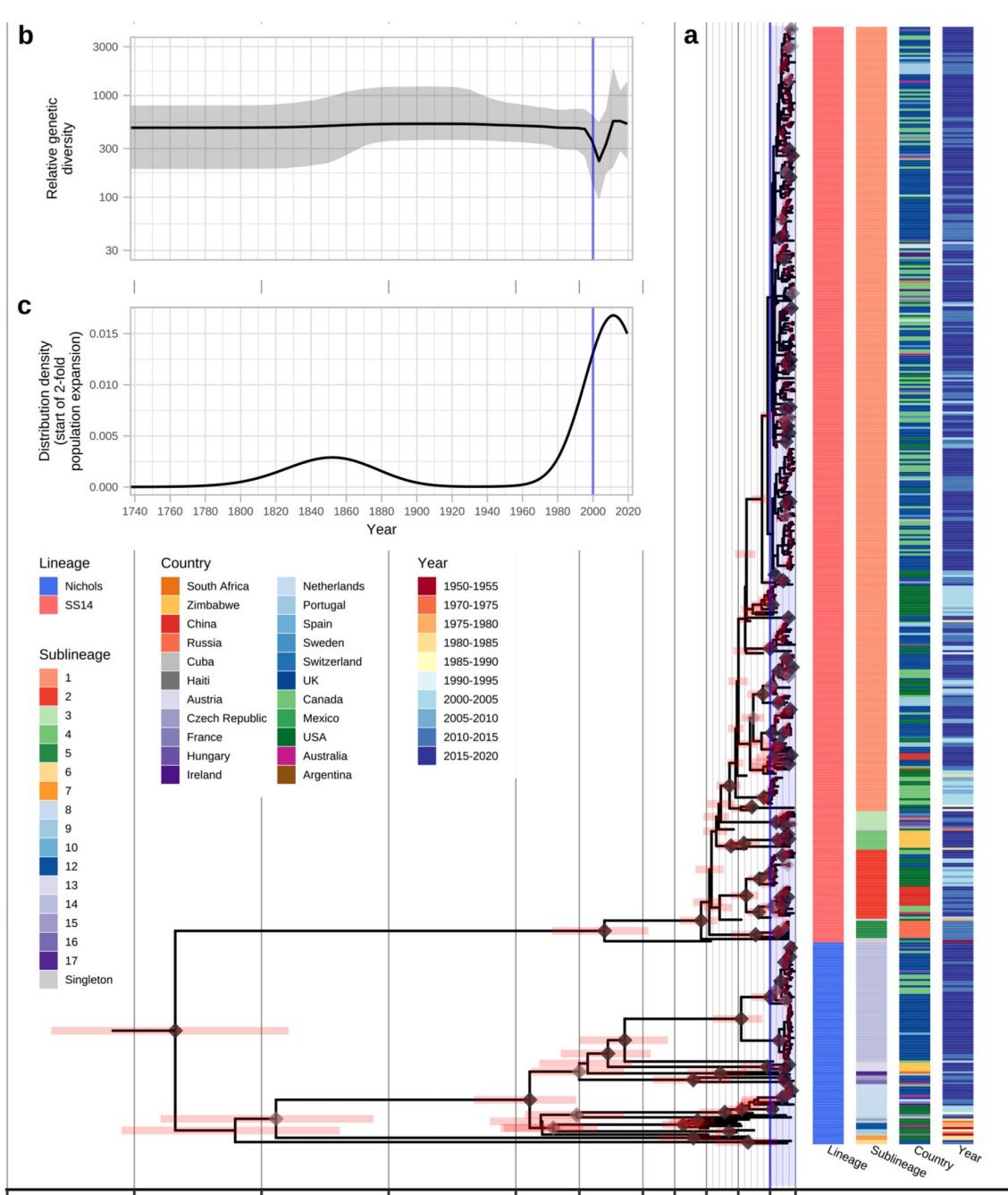

**Figure 3. Bayesian maximum credibility phylogeny of 520 TPA genomes shows population contraction during the 1990s, followed by rapid expansion from the early 2000s onwards.**
A- Time-scaled phylogeny of 520 TPA genomes. Node points are shaded according to posterior support (black 96%, dark grey >91%, light grey >80%). Red bars on nodes indicate 95% Highest Posterior Density intervals. Blue line and shaded area highlight post-2000 expansion of lineages. B- Bayesian Skyline plot of genetic diversity shows small population expansion and contractions during the 19th and 20th Centuries, followed by a sharp decline and rapid re-emergence during the 1990s and 2000s. C- Posterior distribution

of start dates for a 2-fold expansion above skyline mean shows strong support for expansion after 1990 in 68.6% (9263/13503) of trees.

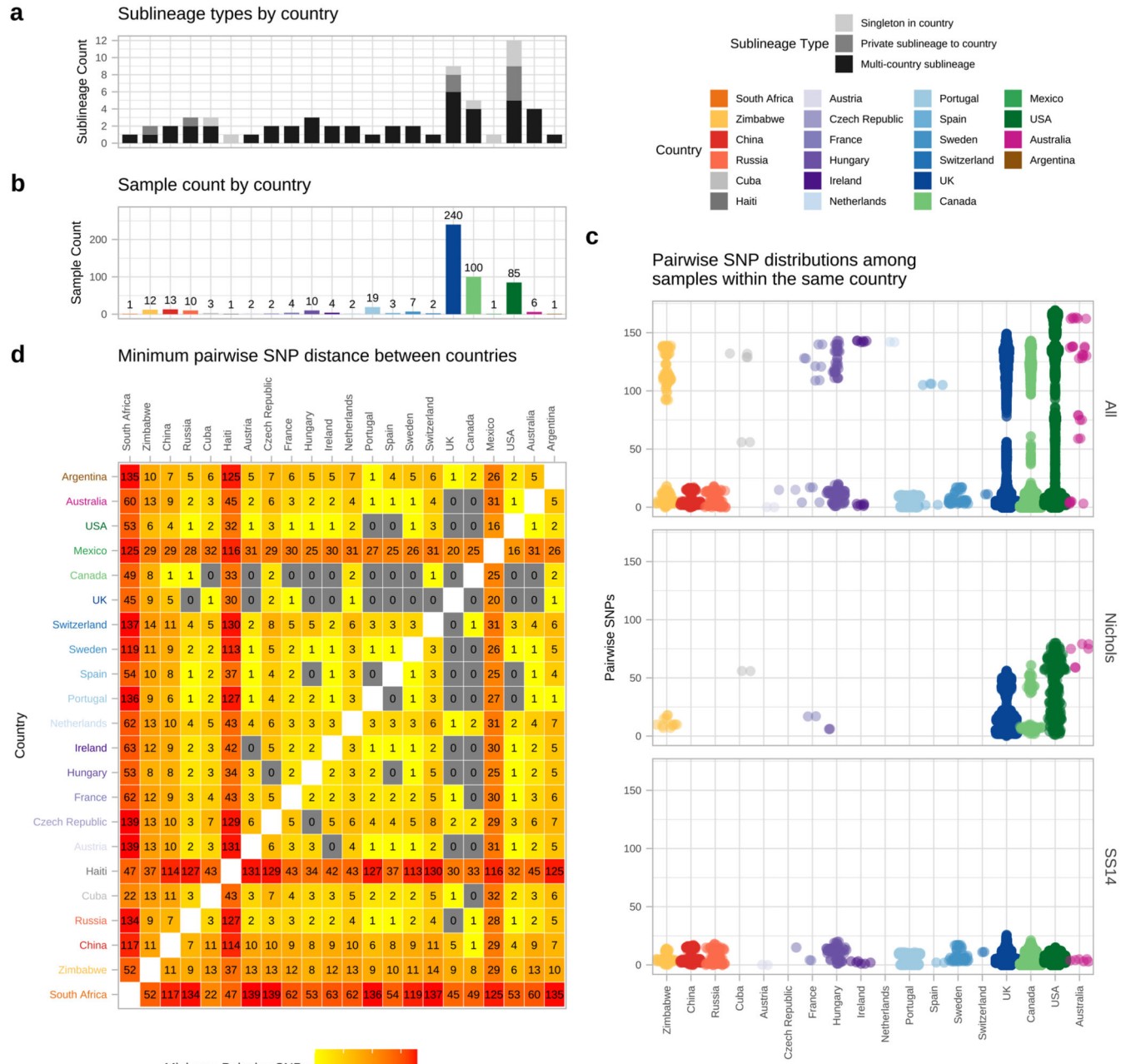

**Figure 4. Substantial sharing of closely related strains within and between countries.**
A- Number of sublineages found per country, classified by sublineage distribution (multi-country=black, private to one country=medium grey, singleton=light grey). B- Total high-quality genomes per country. C- Pairwise comparison of SNP distance distributions from samples in each country (where >1 sample), across all samples and within lineages. D-Minimum pairwise SNPs between samples from different countries. All pairwise SNP comparisons exclude comparisons with same samples. Haiti, South Africa and Mexico appear striking outliers in terms of genetic relatedness (D), but this reflects that the Haiti and

Mexico samples were collected in the 1950s, and we had only a single genome from these countries.

