## [Peer Review File · Nature microbiology]

Peer Review Information

Journal: Nature Microbiology

Manuscript Title: Global phylogeny of *Treponema pallidum* lineages reveals recent expansion and spread of contemporary syphilis

Corresponding author name(s): Mathew Beale

Editorial Notes:

Redactions – published data Parts of this Peer Review File have been redacted as indicated to remove third-party material.

Reviewer Comments & Decisions:

Decision Letter, initial version:

Dear Mat,

Thank you for your patience while your manuscript "Contemporary syphilis is characterised by rapid global spread of pandemic *Treponema pallidum* lineages" was under peer-review at Nature Microbiology. It has now been seen by 4 referees, whose expertise and comments you will find at the end of this email. Although they find your work of some potential interest, they have raised a number of concerns that will need to be addressed before we can consider publication of the work in Nature Microbiology.

In particular, referee #1 says a limitation of the study is the absence of quantum leap change in information content compared to previous studies. This referee also has some concerns over the limited genome coverage and sequencing depth of the genomes. Editorially, we feel this will be important to address. Furthermore, referee #1 is concerned about the choice of the reference genome and about the bootstrap procedures used. Moreover, this referee also has concerns regarding the relatively low number and diversity of South American strains used. Referee #2 suggests to analyse in more detail the two lineages Nichols and SS14. Referee #3 lists several points for improvement of the manuscript, and referee #4 asks how the previously reported periodicity of 8-11 years in the US aligns with the

population dynamics inferences reported from the genomic analyses in this study. Further, this referee suggests to consider the potential role of immune pressure in shaping the population dynamics.

Should further experimental data and text modifications allow you to address these criticisms, we would be happy to look at a revised manuscript.

Please include a data availability statement as a separate section after Methods but before references, under the heading "Data Availability". This section should inform readers about the availability of the data used to support the conclusions of your study. This information includes accession codes to public repositories (data banks for protein, DNA or RNA sequences, microarray, proteomics data etc...), references to source data published alongside the paper, unique identifiers such as URLs to data repository entries, or data set DOIs, and any other statement about data availability. At a minimum, you should include the following statement: "The data that support the findings of this study are available from the corresponding author upon request", mentioning any restrictions on availability. If DOIs are provided, we also strongly encourage including these in the Reference list (authors, title, publisher (repository name), identifier, year). For more guidance on how to write this section please see: <http://www.nature.com/authors/policies/data/data-availability-statements-data-citations.pdf>

* If you have not done so already we suggest that you begin to revise your manuscript so that it conforms to our Article format instructions at <http://www.nature.com/nmicrobiol/info/final->

submission. Refer also to any guidelines provided in this letter.

[REDACTED]

Note: This url links to your confidential homepage and associated information about manuscripts you may have submitted or be reviewing for us. If you wish to forward this e-mail to co-authors, please delete this link to your homepage first.

Nature Microbiology is committed to improving transparency in authorship. As part of our efforts in this direction, we are now requesting that all authors identified as 'corresponding author' on published papers create and link their Open Researcher and Contributor Identifier (ORCID) with their account on the Manuscript Tracking System (MTS), prior to acceptance. This applies to primary research papers only. ORCID helps the scientific community achieve unambiguous attribution of all scholarly contributions. You can create and link your ORCID from the home page of the MTS by clicking on 'Modify my Springer Nature account'. For more information please visit www.springernature.com/orcid.

If you wish to submit a suitably revised manuscript we would hope to receive it within 6 months. If you

cannot send it within this time, please let us know. We will be happy to consider your revision, even if a similar study has been accepted for publication at Nature Microbiology or published elsewhere (up to a maximum of 6 months).

Yours sincerely,
[REDACTED]

Reviewer Expertise:

Referee #1: Syphilis, pathogenomics

Referee #2: Infectious disease evolution

Referee #3: Treponema Genomics/Syphilis

Referee #4: Microbial Genomics/Genomic Epidemiology/Population

Reviewer Comments:

Reviewer #1 (Remarks to the Author):

The Beale et al manuscript is an interesting and rather integrative piece of work dealing with the origin and spread of syphilis worldwide. The dataset, though biased to a large extent, represents the best available collection today, essentially relying on direct whole genome sequencing technology. The analyses implement state of the art algorithms and tools, and seem to be handled the right way. The manuscript might be acceptable in Nature Microbiology, though I still have a certain number of comments and suggestions to improve the manuscript. Another limitation is the absence of quantum leap change in information content compared to the Arora et al paper published in this journal in 2016 and the Beale et al paper published in Nature Communication in 2019.

- One first concern relies on the rather poor coverage and sequencing depth of those genomes, relying on a very specific technology and limited amounts of genetic material (i.e bacterial loads on swabs). I fully understand this limitation, yet the accumulation of N's in the matrices might affect to a certain extent the phylogenetic reconstructions, as well as the Bayesian inferences. This issue could be tested by implementing simulations starting from let say 100 virtual genomes (for example, generated with SLiM, with given mutation rates and demographic scenarios), and then, comparing the outcomes (trees and Beast demogenetics) from the perfect dataset, with datasets where noise proportions are increased to reach the levels observed in the real data. How robust is the signal?

- Is the SS14_V2 reference genome (NC_021508.1) the best choice? According to the authors, not necessarily. Is it not worth to test the mapping on a more modern and representative genome with high coverage?
- Bayesian skylines of both lineages taken independently might shade new light on the demographic scenarios (SS14 versus Nichols).
- The authors tried to define phylogenetic clusters by implementing bootstrap procedures. Such approaches when applied to large datasets and taxa number often provide low bootstrap values due to biases in such procedures and a simplistic binary function. I would recommend using the transfer bootstrap expectation (TBE) developed by Olivier Gascuel, which performs better and relies on a continuous function, to define those groups.
- There is still an ongoing discussion concerning the geographical source of TPA, yet South America leads the race. Unfortunately, this subcontinent has only a few representative strains from a single country (Argentina) and probably from white European patients. This point needs to be discussed. Pushing the argument to its limits, the design looks like a comprehensive study of SARS-CoV-2 without Chinese samples.

Minor points:

- In the introduction (line 87) the authors write, “commonly believed”. There is enough evidence from medical and history books, as well as from novels to change this sentence into “syphilis has caused...”
- Figure 1 and 2 are a bit difficult to read (probably not for stamp collectors).
- Figure 3a. Please name the two major clades on the figure to facilitate the interpretation.
- Figure 3b in the main text looks rather flat and I can hardly see fluctuations prior the first decline in the 1990s. However this is not the case when I scrutinized the supplementary figures 16 and 17. Please fix this.

Reviewer #2 (Remarks to the Author):

The manuscript by Beale et al presents a phylogenomic study of global *Treponema pallidum* that is unrivaled in number or in geographical scope. The methodology is solid and the exposition is clear. The conclusions are well supported by the data. The authors have managed to gather a unique dataset in an

organism for which genomics is more difficult because of the difficulty culturing *T. pallidum* and the need to use direct amplification for whole genomes. While the phylogenomic (dynamic) analysis is solid, the manuscript would have been stronger if the authors could have looked deeper into the two lineages (Nichols and SS14). What are the major genes/changes that separate these two lineages? Do they point to any biological differences? Can the authors expand their comments on why the two lineages seem to have diversified at different times despite being present contemporaneously? More commentary (or even some speculation) on why these two lineages may be equally successful would be helpful.

A few minor comments:

Line 109: insert “understanding of *the Nichols lineage”

Line 123: The term panmictic is confusing to me here. I believe that the authors mean that lineages are spread widely across the globe even at the sublineage level. When I think of a panmictic bacterial species I think of one that has so much horizontal transfer that it erases phylogenetic signal.

Line 145: The lack of Latin American strains should be noted as a limitation in the limitations section.

Did the authors calculate an effective population size with their Bayesian Skyline plot? This would be interesting to see.

Reviewer #3 (Remarks to the Author):

Review Beale et al.

Beale et al. Present an extensive analysis of *Treponema pallidum* (TPA) genomes, many of which they have obtained in this project. Genomic epidemiology has become to the scientific and social spotlight in the last year and this manuscript clearly shows much of its power with a particularly difficult organism. Contrarily to viruses and most other bacterial species to which genomic epidemiology is now routinely applied, TPA has been considered a monomorphic species with an additional technical difficulty in obtaining complete genome sequences. Recent developments in enrichment and other advances open a new window of possibilities for genome analysis of TPA and Beale et al.’s manuscript provides a first glimpse of the current genomic variation in this pathogen. The most remarkable findings are the documentation of the simultaneous circulation, at a worldwide scale, of the two main lineages of TPA, Nichols and SS14, with ongoing diversification in sublineages that apparently is not related to adaptation to azithromycin, despite the detection of many samples that carry the two mutations known to confer resistance to this antibiotic. Additionally, the phylodynamic analyses reveal different periods of expansion and contraction in the incidence of infections that closely match known changes in the

epidemiology of syphilis, thus allowing a better understanding of the relationships between epidemiological dynamics and genome features.

The analyses performed have employed some of the most sophisticated and up-to-date techniques currently available. To overcome some of the difficulties in the analysis of the complete dataset they have used several strategies that reinforce their results and the conclusions derived from them. In this respect, the authors have made an excellent job that many will be able to follow for analysing similarly difficult data sets. I think that the manuscript is innovative, of high quality, and represents a significant advancement in the application of genomic analysis of pathogenic bacteria. Consequently, I recommend its publication in Nature Microbiology, although there are some minor points that will likely improve it and that I detail next.

In Figure 2, the ladder-like pattern in the ML trees of the SS14 sublineages will likely be corrected by using the “collapse zero-length branches” option in IQTREE (-czb).

I miss some additional information on the epidemiology of syphilis at least for those areas that are well covered and a comparison with the changes observed in the corresponding phylodynamic analyses beyond the general

The identification of transmission of identical genomes, even among different countries, represents a serious problem for the epidemiological reconstruction of transmissions but also for forensic microbiology.

The authors mention (lines 432-435) possible convergent adaptation of samples from different lineages to the rabbit host through passages. Have they analysed which changes may have driven these adaptations?

I miss some discussion on the separation between the Nichols and SS14 lineages of TPA.

(line 190) the 5% BS for considering sublineages is a very low value. What is the justification for using such a low value? How many sublineages will be retained should a higher value be considered, such as 50% or even higher?

Why is mapping coverage 48.6X with simulated reads from RefSeq closed genomes (Supplemental Data 1)?

Lines 569-571: How was recombination inferred in the 19 regions? Supp. Data 2 shows regions inferred by Gubbins, but this is not described in the main text. The references cited in the sentence include 12 recombinant loci (refs. 3,6) or use Gubbins with a different dataset (ref. 4) which identified 23 regions to mask (12 as hypervariable and 11 with Gubbins).

Line 592. About the 10 SNP threshold to determine clusters, How was this value chosen? In light of the sites removed, is this proposed as a threshold to separate isolates into different clusters or should it be adjusted for each dataset?

I would like some discussion on the inferred molecular clock rate (line 618) for the core genome analyses of TPA sequences and what is the likely impact of changes in the accessory genome.

Reviewer #4 (Remarks to the Author):

This is a very well done and clearly written manuscript that advances understanding of the history and epidemiology of syphilis and the population genomics of the causative organism *Treponema pallidum* subsp *pallidum*. It reports findings that will be of broad interest.

My main question centers on the comparison to the syphilis incidence patterns in <https://pubmed.ncbi.nlm.nih.gov/15674292/>. In that paper, the authors report a periodicity of 8-11 years in the US. How does that align with the population dynamics inferences reported from the genomic analyses? As the dataset is primarily from 2010-20, can it be used to address this question, or could the data specifically from the US and Canada, the two places with isolates spanning two or more decades, help provide an answer? Are you confident in the population dynamics preceding ~2000? Similarly, I'd be interested in the authors' thoughts on the role of immune pressure in shaping the population dynamics.

Relatedly, are the regions of recombination in genes known or thought to be antigenic, as suggested in <https://www.frontiersin.org/articles/10.3389/fmicb.2019.01691/full?>

Minor points.

Lines 122-3. What does "essentially panmictic" mean? Is this a comment about randomness of association between loci within the genome (a la <https://pubmed.ncbi.nlm.nih.gov/8506277/>) or about the geographic distribution of lineages?

Line 251. Caution about calling something extinct, given the level of sampling

Gubbins: how did you decide on 20 iterations? If I recall correctly, the default is 5.

Supplementary Figure 1.

Hard to tell many of the European countries apart, as the color hues are so similar. If you want readers to be able to identify countries in this plot, I think you'll have to come up with another scheme. One idea (not obligatory, just a suggestion): put lineage on the inner annotation ring and then label countries with color and a letter (like the numbering of lineages in supplemental figure 2).

Supplementary Figure 4. I believe the title of the middle annotation column is missing a letter.

Supplementary Figure 6. Quibble: the Nichols reference genomes aren't unrelated, they're in a separate clade.

Supplementary Figure 7. Unclear how to read the plot on the right. Does the black/gray mean the same as in the key to the left? If black is 'direct from clinical sample' what does that mean for, say, lineage 8?

Author Rebuttal to Initial comments

Reviewer Expertise:

Referee #1: Syphilis, pathogenomics

Referee #2: Infectious disease evolution

Referee #3: Treponema Genomics/Syphilis

Referee #4: Microbial Genomics/Genomic Epidemiology/Population

Reviewer Comments:

Reviewer #1 (Remarks to the Author):

The Beale et al manuscript is an interesting and rather integrative piece of work dealing with the origin and spread of syphilis worldwide. The dataset, though biased to a large extent, represents the best available collection today, essentially relying on direct whole genome sequencing technology. The analyses implement state of the art algorithms and tools, and seem to be handled the right way. The manuscript might be acceptable in Nature Microbiology, though I still have a certain number of comments and suggestions to improve the manuscript. Another limitation is the absence of quantum leap change in information content compared to the Arora et al paper published in this journal in 2016 and the Beale et al paper published in Nature Communication in 2019.

We thank the reviewer for their positive comments. Please see our discussion above regarding our view on the relative advance this work represents.

- One first concern relies on the rather poor coverage and sequencing depth of those genomes, relying on a very specific technology and limited amounts of genetic material (i.e bacterial loads on swabs). I fully understand this limitation, yet the accumulation of N's in the matrices might affect to a certain extent the phylogenetic reconstructions, as well as the Bayesian inferences. This issue could be tested by implementing simulations starting from let say 100 virtual genomes (for example, generated with SLiM, with given mutation rates and demographic scenarios), and then, comparing the outcomes (trees and Beast demogenetics) from the perfect dataset, with datasets where noise proportions are increased to

reach the levels observed in the real data. How robust is the signal?

The reviewer is correct that obtaining high quality genome data from primary *Treponema* clinical samples is extremely technically challenging, and although we have pioneered these approaches for several human pathogens it can indeed result in a high proportion of N sites within the multiple sequence alignments. In appreciation of this we used only genomes with <25% of sites masked to N in our fine scale analyses. It is notable that the same basic tree topology is obtained when including less stringently filtered sites, such as Supplementary Figure 1, based on genomes with <75% of sites masked to N.

Moreover, we are not clear what the reviewer means by a 'perfect genome' or dataset, since the TPA genome has regions known to be problematic for phylogenetic analysis due to being highly repetitive, hypervariable and/or recombining. It is standard practice in microbial genomics to mask these sites for phylogenetics, and in our analysis these accounted for up to 4.7% of genomic sites. Within the 528 genomes used for finescale analysis, 301 (58%) genomes had <5% of sites masked to N – see cumulative plot below.

{REDACTED}

Therefore, to address this point, we have performed maximum likelihood phylogenetic reconstruction of these 301 genomes, which shows conservation of our sublineage clusters (see plot below), but excludes the two outlying sublineage 6 samples (10% and 13% of sites masked to N respectively) and sublineages 13 and 15.

We now include a tanglegram (Supplementary Figure 17) comparing the ultrametric trees of the full 528 genomes with the new 301 genome tree, which shows equivalent topology. Note that as described in the manuscript, sublineage 1 contains extremely low genetic diversity, resulting in low bootstrap support within this clade, and as expected, this leads to some minor topological rearrangements associated with this sublineage. This is not because strains are not closely related, but rather because sublineage 1 strains are so closely related that the topology shifts with each bootstrap.

We find equivalent clustering and topology whether requiring <5% or <25% N's within the multiple sequence alignment, so in answer to your question, our conclusions are robust and any minor changes in the topology of the tree are not biologically meaningful.

- Is the SS14_V2 reference genome (NC_021508.1) the best choice? According to the authors, not necessarily. Is it not worth to test the mapping on a more modern and representative genome with high coverage?

T. pallidum genomes are almost entirely syntenic, with no accessory genome (the main reason why reference genome selection can be so crucial in other bacterial genomics) and moreover, the maximum SNP distance between any two genomes in our entire study was 175, so the reference genome selection does not have a substantive effect on our analysis. To address this, we have repeated the mapping and maximum likelihood tree and, as you can see below when using the Nichols_v2 reference genome, whilst there are very minor differences in the topology, the sublineage relationships are preserved (as discussed elsewhere, sublineage 1 has low diversity, and the inconsistent phylogenetic reconstruction is expected in this part of the tree). To address your concern we have included the tanglegram as a Supplementary Figure 18). It is important to note that the conclusions of this manuscript remain unchanged.

- Bayesian skylines of both lineages taken independently might shade new light on the demographic scenarios (SS14 versus Nichols).

Thank you for this suggestion. Our analysis was fundamentally about recent evolutionary history, focussing on the sublineages, rather than historical SS14 and Nichols lineages. Within both the SS14 and Nichols clades, it is clear that there are sublineages which have expanded rapidly, whilst others have not (within the constraints of our sampling framework). Therefore, we focussed on these dynamics in our analyses.

However, we have now performed the equivalent Bayesian Skyline analyses of the SS14 and Nichols clades and this shows that there are indeed different phylodynamics between SS14 and Nichols. Our new skyline analysis (Supplementary Figure 11A) shows that the major collapse in relative genetic

diversity observed in the full dataset around 2000 (Figure 3B) appears to have occurred predominantly in the Nichols lineage, and this coincides with rapid expansion of SS14 lineage – further supporting evidence of historic lineage replacement. Importantly, BEAST’s ‘lineages over time’ plot (Supplementary Figure 11B), which examines the cumulative generation of new branches, indicates that both SS14 and Nichols continued to diversify, but Nichols was doing so much more slowly until around 2010, at which point there was a change in the steepness of the slope. This is consistent with the rising credible intervals shown at that time for Nichols in the Bayesian Skyline plot, as well as the large clonal expansion we observed for the Nichols sublineage 14 in multiple countries.

- The authors tried to define phylogenetic clusters by implementing bootstrap procedures. Such approaches when applied to large datasets and taxa number often provide low bootstrap values due to biases in such procedures and a simplistic binary function. I would recommend using the transfer bootstrap expectation (TBE) developed by Olivier Gascuel, which performs better and relies on a continuous function, to define those groups.

Thank you for the suggestion. However, the TBE method was specifically designed for resolving issues with Felsenstein’s bootstrap. For our maximum likelihood phylogenetic analyses, we used the UFboot method of Minh and colleagues (MBE, 2013), which provides its own solution to the same issues the reviewer highlights, is also optimised for large datasets and already has a similar bias correction method included. These two methods are incompatible in IQ-Tree, as TBE is redundant (see discussion here <https://groups.google.com/g/iqtree/c/ULJmzhRr4zA>).

For our clustering, although we also made use of the standard non-parametric bootstrap method, we used this only as a way to generate a sample of trees. For interest, we also did the same thing by subsampling from the posterior distribution of PhyloBayes trees with equivalent results, and therefore opted to use the IQ-Tree version in the manuscript. For rPinecone clustering, we then operated only on the trees generated, not on the calculations inferred, and therefore TBE would not be appropriate in this context.

- There is still an ongoing discussion concerning the geographical source of TPA, yet South America leads the race. Unfortunately, this subcontinent has only a few representative strains from a single country (Argentina) and probably from white European patients. This point needs to be discussed. Pushing the argument to its limits, the design looks like a comprehensive study of SARS-CoV-2 without Chinese samples.

The Reviewer will be aware that the community is still deeply divided about the “Columbian Hypothesis”, indeed Majander (Current Biology, 2020) and Giffin (Scientific Reports, 2020) provide a compelling argument against a Latin American origin (or the so-called “Columbian Hypothesis) for *T. pallidum*.

However, as we have clearly laid out in the manuscript our study was not focussed on historical events of

~500 years ago, but on recent evolutionary history. We found evidence of potential sublineage extinctions. We also show global radiation of common contemporary lineages, and because of this we would expect to find most common sublineages in our global collection. Ancestral strains that were not globally distributed (and had not become extinct) would likely be rare, and require deep sampling of local contemporary populations or ancient DNA sequencing of historical remains to detect. This was well beyond the scope of the current study.

Minor points:

- In the introduction (line 87) the authors write, “commonly believed”. There is enough evidence from medical and history books, as well as from novels to change this sentence into “syphilis has caused...”

The recent ancient genome findings of Majander et al (Current Biology, 2020), Giffin (Scientific Reports, 2020), and our discussion of the former manuscript (Beale and Lukehart, Current Biology, 2020) illustrate that this still an open question. As we state in our commentary there:

“Historical evidence — including reports of syphilis and skeletal evidence of Treponema infection in pre-Columbian archaeological remains in Europe [5] — is not definitive: historical documents may confuse syphilis with other diseases; bone abnormalities cannot discriminate among treponematoses; and ancient treponemal infections have been studied little outside of Europe and North America [2]. Furthermore, even the diagnosis of treponematoses in contemporary patients can be uncertain due to overlapping clinical manifestations and bacterial aetiologies.”

Indeed, venereal syphilis and gonorrhoea were only recognised as separate conditions within the last two hundred years, and we now know that *Treponema pallidum* subspecies *pertenue* was also present in Europe at around the same time period.

- Figure 1 and 2 are a bit difficult to read (probably not for stamp collectors).

When describing the largest and most diverse group of TPA genomes ever published we acknowledge there are challenges in legibility. To aid this we included expanded versions of the Nichols and SS14 phylogenies provided (please see Supplementary Figures 4, 5 and 6).

- Figure 3a. Please name the two major clades on the figure to facilitate the interpretation.

Done

- Figure 3b in the main text looks rather flat and I can hardly see fluctuations prior the first decline in the 1990s. However this is not the case when I scrutinized the supplementary figures 16 and 17. Please fix this.

In Figure 3B, the Bayesian Skyline plot is based on the full 520 genome dataset, and while the confidence

intervals fluctuate, the median genetic diversity is reasonably stable prior to the 1990s. This does indeed appear slightly different to the medians from supplementary figures 16 and 17 (now relabelled as Supplementary Figures 20 and 21 in this revision) which appear to fluctuate slightly through the late 19th and 20th Centuries – both of those Skyline analyses are based on subsampled datasets of 138 and 168 genomes respectively. None of the three Skyline plots is incorrect, but the one shown in Figure 3B is derived from a larger, more complete dataset, and is thus the most robust.

Moreover, the focus of our study was on the recent evolutionary history of TPA (which is consistent between all three figures), not on the historical context to which this question pertains.

Reviewer #2 (Remarks to the Author):

The manuscript by Beale et al presents a phylogenomic study of global *Treponema pallidum* that is unrivaled in number or in geographical scope. The methodology is solid and the exposition is clear. The conclusions are well supported by the data. The authors have managed to gather a unique dataset in an organism for which genomics is more difficult because of the difficulty culturing *T. pallidum* and the need to use direct amplification for whole genomes. While the phylogenomic (dynamic) analysis is solid, the manuscript would have been stronger if the authors could have looked deeper into the two lineages (Nichols and SS14). What are the major genes/changes that separate these two lineages? Do they point to any biological differences?

We thank the review for their positive comments.

A comparative analysis of SS14 and Nichols was performed in Matejková (2008) on initial publication of the SS14 genome, and again in Pětrošová (2013) when it was resequenced, and this has also been revisited on a number of occasions. There were no obvious candidates that might affect biology, and any SNPs inferred from a new analysis would need to be associated with function – this would require a purpose-designed study aimed at exploring this function, and is therefore well outside the scope the current manuscript. Nevertheless, we have now used our phylogenetic framework to perform ancestral reconstruction and determine the SNPs (and their coding effects) that separate the inferred MRCA of contemporary SS14 from that of the contemporary Nichols expansions in our multiple sequence alignment (see Supplementary Data 3 and Supplementary Figure 16).

Can the authors expand their comments on why the two lineages seem to have diversified at different times despite being present contemporaneously? More commentary (or even some speculation) on why these two lineages may be equally successful would be helpful.

Our data support investigation of the recent evolutionary history of TPA, and therefore this the focus of

our analysis. However, we consider it interesting that while we find multiple branching events along the Nichols lineage, including those leading to sublineages 6 and 7, the first evidence of SS14 branching (aside from Mexico A) occurs in the 1950s (consistent with Arora 2016). This begs the question 'where was SS14 before this', since if it were circulating in the general population we would expect to find extant descendent lineages from ancestral nodes. Since TP is an obligate human/primate pathogen which must exist in a host, we can speculate that SS14 may have previously been present in an isolated population, and that perhaps coming into contact with different populations lead to its dissemination. Our lack of historical samples makes this highly speculative, but the dating of dissemination in the 1950s could imply international spread following the Second World War.

A few minor comments:

Line 109: insert "understanding of *the Nichols lineage"

Done

Line 123: The term panmictic is confusing to me here. I believe that the authors mean that lineages are spread widely across the globe even at the sublineage level. When I think of a panmictic bacterial species I think of one that has so much horizontal transfer that it erases phylogenetic signal.

It was our intention to invoke the lack of geographical signal, but we agree that this term can be interpreted differently, so have changed this to 'genetically homogenous'.

Line 145: The lack of Latin American strains should be noted as a limitation in the limitations section.

This section of the Results highlighted new genomes, rather than all the genomes in the study. As shown in Figure 1, we included genomes from Argentina, Cuba, Mexico and Haiti (all Latin American countries), and we now highlight this on line 141. Like Africa and Asia, the sampling from South America, Central America and the Caribbean is indeed limited in our collection, and we addressed the limitations of the collection on lines 497-498, but have now rearranged this to further emphasise the limited sampling in these regions.

Did the authors calculate an effective population size with their Bayesian Skyline plot? This would be interesting to see.

The default output of the Bayesian Skyline plot is effective population size multiplied by generation time, so this can be converted to effective population size if the generation time is known. However, we were concerned that using this term for a broad audience could lead readers to misinterpret our analysis as directly inferring global population sizes. We therefore chose to refer to this as "relative genetic diversity", a term which is now widely used within the field when referring to pathogen phylodynamics (e.g. Rambaut et al 2008, Parag et al 2020).

Reviewer #3 (Remarks to the Author):

Beale et al. Present an extensive analysis of *Treponema pallidum* (TPA) genomes, many of which they have obtained in this project. Genomic epidemiology has become to the scientific and social spotlight in the last year and this manuscript clearly shows much of its power with a particularly difficult organism. Contrarily to viruses and most other bacterial species to which genomic epidemiology is now routinely applied, TPA has been considered a monomorphic species with an additional technical difficulty in obtaining complete genome sequences. Recent developments in enrichment and other advances open a new window of possibilities for genome analysis of TPA and Beale et al.'s manuscript provides a first glimpse of the current genomic variation in this pathogen. The most remarkable findings are the documentation of the simultaneous circulation, at a worldwide scale, of the two main lineages of TPA, Nichols and SS14, with ongoing diversification in sublineages that apparently is not related to adaptation to azithromycin, despite the detection of many samples that carry the two mutations known to confer resistance to this antibiotic. Additionally, the phylodynamic analyses reveal different periods of expansion and contraction in the incidence of infections that closely match known changes in the epidemiology of syphilis, thus allowing a better understanding of the relationships between epidemiological dynamics and genome features. The analyses performed have employed some of the most sophisticated and up-to-date techniques currently available. To overcome some of the difficulties in the analysis of the complete dataset they have used several strategies that reinforce their results and the conclusions derived from them. In this respect, the authors have made an excellent job that many will be able to follow for analysing similarly difficult data sets. I think that the manuscript is innovative, of high quality, and represents a significant advancement in the application of genomic analysis of pathogenic bacteria. Consequently, I recommend its publication in *Nature Microbiology*, although there are some minor points that will likely improve it and that I detail next.

We thank the review for their very positive and kind comments.

In Figure 2, the ladder-like pattern in the ML trees of the SS14 sublineages will likely be corrected by using the “collapse zero-length branches” option in IQTREE (-czb).

Thank you for this – we agree that the ladder-like appearance for some of the clonal expansion was problematic, and have applied the suggested solution to the underlying trees and affected figures. Note that doing so has affected the placement of sublineage 6, which diverges from a node very close to the root of all TPA – in the revised midpoint-rooted maximum likelihood phylogeny sublineage 6 is now placed on the SS14 side of the tree, whilst it remains on the Nichols side for the BEAST trees. This has necessitated some rewording of the text, and revision of most figures.

I miss some additional information on the epidemiology of syphilis at least for those areas that are well covered and a comparison with the changes observed in the corresponding phylodynamic analyses beyond the general

We were not clear exactly what epidemiology the reviewer refers to, but if this query refers to infection rates, we utilised published rates for British Columbia (Canada) and England (UK) as a comparator to the genomic data from the best sampled countries in Supplementary Figure 13A (relabelled 14A in this revised version). If this refers to epidemiological patterns associated with sublineages in our study, we considered this, but because our study is the result of samples collected from a large group of laboratories, each with different patient data collection practices (and public release approval processes), the overall completeness of the metadata was very inconsistent, and there were real dangers that any interpretations made here would be subject to group-specific biases. Where we have consistent high quality patient metadata, e.g. from the UK, this warrants a much more forensic analysis than would be possible here, and will be presented in a separate paper.

The identification of transmission of identical genomes, even among different countries, represents a serious problem for the epidemiological reconstruction of transmissions but also for forensic microbiology.

Yes, we strongly agree.

The authors mention (lines 432-435) possible convergent adaptation of samples from different lineages to the rabbit host through passages. Have they analysed which changes may have driven these adaptations?

We have not. However, this is a great idea for a future study – thank you.

I miss some discussion on the separation between the Nichols and SS14 lineages of TPA.

The dating of separation between the two lineages has previously been discussed in detail elsewhere (e.g. Arora 2016, Majander 2020), including by us (Beale, 2019). Here, we focussed our analysis on the recent evolutionary dynamics of contemporary transmission.

(line 190) the 5% BS for considering sublineages is a very low value. What is the justification for using such a low value? How many sublineages will be retained should a higher value be considered, such as 50% or even higher?

We evaluated a number of different approaches for defining clusters in *T. pallidum*, including hierBAPS and PopPUNK. However, the clonality of TPA genomes is such that these methods cannot resolve sublineages beyond a certain level. rPinecone is a method that infers SNP distances between samples from common ancestral nodes, and uses this to form clusters. This is theoretically superior to simple SNP thresholds or patristic distance, because it accounts for evolutionary directionality. However, it is highly dependent on tree topology. We initially performed our rPinecone clustering using the maximum likelihood phylogeny, as intended by the authors of the package, but we found that the areas of low bootstrap support in our tree resulted in minor topological changes between our ML and Bayesian trees

for SS14-lineage, resulting in breaking up of clusters.

To overcome this and to find topologically consistent clusters, we therefore attempted to provide a measure of support with our bootstrapping approach. However, the issue with bootstrapping is that it means subsampling a dataset with already low diversity. We illustrated this in Supplementary Figure 3, where we evaluated different hierarchical clustering thresholds for the rPinecone approach. As Supplementary Figure 3 shows, when requiring 50% of trees to be part of a cluster, we find exactly the same Nichols sublineages, but sublineage 1 (and to a far lesser degree sublineage 4) in SS14- lineage is fragmented. This is not because strains are not closely related, but rather because they are so closely related that the topology shifts with each bootstrap. The larger the proportion of bootstrapped trees we require for clustering, the more fragmented these clusters become (e.g. requiring 95% of trees leads to total collapse of grouping for sublineage 1). The samples we grouped into sublineage 1 for our final analysis represent the coarsest clusters, so we are being conservative with this approach – we appreciate this is counterintuitive and may not have been well communicated in the manuscript text, and have amended the text on lines 188.

Why is mapping coverage 48.6X with simulated reads from RefSeq closed genomes (Supplemental Data 1)?

We simulated 50X 150bp PE reads from the closed genomes, and these were then mapped back to the reference for our analysis pipeline – because repetitive/masked regions do not map, we would not expect mapping coverage to perfectly match 50x, but the resulting mean genome coverage was still very close to 50X, as the reviewer points out. We have now clarified in the methods (line 568) that we simulated reads at 50X, since this was not originally mentioned.

Lines 569-571: How was recombination inferred in the 19 regions? Supp. Data 2 shows regions inferred by Gubbins, but this is not described in the main text. The references cited in the sentence include 12 recombinant loci (refs. 3,6) or use Gubbins with a different dataset (ref. 4) which identified 23 regions to mask (12 as hypervariable and 11 with Gubbins).

We used two approaches for masking regions of uncertainty. Firstly, we masked 19 regions known to be problematic in phylogenetic analyses. For this approach, we masked the same 12 Tpr genes (Tpr A-L) and the two highly repetitive genes (arp, TPANIC_0470), as we did in Beale 2019 (originally influenced by Arora 2016). Majander and colleagues (2020) subsequently showed that masking the 5 FadL homologs was beneficial for resolving phylogenetic signal, so we also masked those in our analysis, making a total of 19. These regions were masked from all genomes at the point of mapping, and again after pseudosequence generation. This was described in our Methods (now on lines 571- 574 and 582-588).

Secondly, we ran Gubbins to detect additional putative recombinogenic regions (described on lines 598-602), and this identified 19 further regions for exclusion (but only in a subset of samples). We did not make use of our previous Gubbins inferences from Beale 2019 in this approach, preferring to perform

a *de novo* analysis with this larger dataset (although most of the same regions were identified). This left us with a total of 38 regions with some level of masking, but it is worth noting that some of the regions identified by Gubbins in different parts of the tree are overlapping (see Supplementary Data 2), so this affects the total number of sites affected. Combined, these intentionally masked sites account for a maximum of 4.7% of sites (see Methods, line 602).

Line 592. About the 10 SNP threshold to determine clusters, How was this value chosen? In light of the sites removed, is this proposed as a threshold to separate isolates into different clusters or should it be adjusted for each dataset?

It is important to recognise that there is no 'right way' to cluster genomes, and any method contains an element of subjectivity or inconsistency. The 10 SNP threshold was originally used in our 2019 Nature Communications manuscript, and provided a systematic method derived directly from the phylogeny to describe the patterns we observed, enabling useful insights into the groupings and evolutionary dynamics of the dataset for that study. For consistency, we chose the same cluster threshold here. However, we do not define a formal typing method for TPA, and it would be for individual investigators to select an appropriate clustering method for different datasets.

As part of the rPinecone workflow, joint ancestral reconstruction is performed on the tree using the multiple sequence alignment, meaning that the identity of sites that might have been masked to N in the sequence at the tip will be inferred from the tree topology at the ancestral nodes. Clustering for rPinecone is based on the SNP distance from the ancestral nodes using these reconstructed sequences. Therefore rPinecone is not strongly affected by sites that are masked to N, with the caveat that it assumes the tree topology previously inferred is correct (see our comments earlier about this), and that if sites masked to N hide novel diversity private to the individual, that might increase the distance from the ancestral node.

I would like some discussion on the inferred molecular clock rate (line 618) for the core genome analyses of TPA sequences and what is the likely impact of changes in the accessory genome.

Treponema pallidum genomes are almost entirely syntenic, and although we use the term 'core genome' in our manuscript, this is to refer to 'core sites', as opposed to 'masked sites'. There is no 'accessory genome' as it would usually be described in bacterial genomes. The masked sites (effectively accessory here) are in many cases known to be recombinogenic (Arora 2016, Beale 2019, Majander 2020), so fall outside the clonal frame of the TPA genome, and are therefore not phylogenetically consistent (but would likely give faster molecular clock rates if included). We have now highlighted that this rate is consistent with Arora 2016 and Beale 2019, and added an additional point about the likely effect of including sites outside the clonal frame on lines 551-554.

Reviewer #4 (Remarks to the Author):

This is a very well done and clearly written manuscript that advances understanding of the history and epidemiology of syphilis and the population genomics of the causative organism *Treponema pallidum* subsp *pallidum*. It reports findings that will be of broad interest.

Thank you for the positive comments.

My main question centers on the comparison to the syphilis incidence patterns in <https://pubmed.ncbi.nlm.nih.gov/15674292/>. [pubmed.ncbi.nlm.nih.gov] In that paper, the authors report a periodicity of 8-11 years in the US. How does that align with the population dynamics inferences reported from the genomic analyses? As the dataset is primarily from 2010-20, can it be used to address this question, or could the data specifically from the US and Canada, the two places with isolates spanning two or more decades, help provide an answer?

Thank you for this really interesting question. We have ourselves considered the work of Grassly and colleagues, including how to integrate their inferred model of periodicity in the US with our own inferences. Whilst noting that we are not ourselves epidemiological modellers, we fully recognise the value of these approaches. However, the Grassly model was derived from aggregated national data from the US, and did not account for known shifts in demography for race/ethnicity and sexual orientation (e.g. the shift to crack-cocaine users during the 1990s that occurred in the US but not elsewhere). This contrasts with our comparatively modest sample, which contains samples from different parts of the US but is dominated by those from Seattle. As the reviewer points out, the majority of samples in our dataset were collected after 2000, particularly after 2010, and this limits our ability to test or observe the 8-10 year cyclic behaviour described by Grassly et al over a longer and earlier time period between 1960-1993. Moreover, our observations are based on the global phylogeny, and it could be that observations made in a single country do not have measurable impact on the global population dynamics.

That said, we do see evidence of novel sublineages rising in discrete populations, e.g. sublineage 14, which appeared contemporaneously in both Canada and the UK, as well as other lineages seemingly declining or becoming extinct (such as the 'Nichols reference cluster'). These observations could potentially be interpreted as linked to population size cycling, but more comprehensive longitudinal sampling and sequencing of populations is needed to address this question.

Are you confident in the population dynamics preceding ~2000?

The Bayesian Skyline analysis relies on inferred dates of ancestral nodes in the phylogeny. As such, we can only infer population dynamics for lineages that exist in contemporary populations. As we describe in our manuscript, there is some suggestion that sublineages may decline or become extinct – if there are

no extant descendants of earlier lineages, we would not be able to make inferences about them. This is a limitation we alluded to in our Discussion on lines 503-507, but we now explicitly discuss this issue on line 508-510 to make this clearer.

Similarly, I'd be interested in the authors' thoughts on the role of immune pressure in shaping the population dynamics.

Although immune pressure is one potential hypothesis for the observed dynamics, this is a difficult question to address from our current data. It is likely that immune pressure would be seen first and primarily within a given local sexual network (e.g. AA or MSM), not in aggregated national data as used by Grassly, or dispersed national and international data as in parts of our dataset. Thus the MSM samples from Seattle or Vancouver would be a logical place to look and, in fact, data from both Seattle (Marra 2021, CID doi: 10.1093/cid/ciab287) and Antwerp (Kenyon 2018, BMC Infectious Diseases) support that reinfections are more likely to be asymptomatic in persons with multiple previous syphilis episodes. However, we had insufficient longitudinal sampling to address this question for Seattle, insufficient metadata for Vancouver, and only one sample from Antwerp.

Moreover, were immunological pressure to be important, the likely candidate genes would be the highly recombinogenic antigenic proteins (e.g. Tpr's) that are excluded from the clonal frame for phylogenetics. Therefore, investigation of this will require both a purpose designed dataset and dedicated analyses focusing on these proteins, and is thus outside the scope of the current work. However, we now discuss these important points in our Discussion on lines 479-484.

Relatedly, are the regions of recombination in genes known or thought to be antigenic, as suggested in <https://www.frontiersin.org/articles/10.3389/fmicb.2019.01691/full?> [frontiersin.org]

Yes, most of the major immunologically-relevant genes in Tp are also recombinogenic, and would have been excluded from our phylogenetic analysis.

Minor points.

Lines 122-3. What does "essentially panmictic" mean? Is this a comment about randomness of association between loci within the genome (a la <https://pubmed.ncbi.nlm.nih.gov/8506277/> [pubmed.ncbi.nlm.nih.gov]) or about the geographic distribution of lineages?

Following comments from Reviewer 2 and yourself, 'panmictic' has now been changed to say "genetically homogenous".

Line 251. Caution about calling something extinct, given the level of sampling.

We agree, and on line 252 we had caveated this with "within our sampling framework", and we now also make clear that this could be a "decline to rarity" rather than necessarily an extinction (line 253).

Gubbins: how did you decide on 20 iterations? If I recall correctly, the default is 5.

The reviewer is almost correct, in that the default for Gubbins is to run a *maximum* of 5 iterations – after each iteration, Gubbins evaluates the evidence for further recombination through comparing the consistency of previous rounds, before deciding whether to perform a further iteration. In the process of being thorough, we added more optional iterations, and Gubbins continued to run for the full 20 iterations specified, suggesting there is still some unrecognised signal within the dataset.

Supplementary Figure 1. Hard to tell many of the European countries apart, as the color hues are so similar. If you want readers to be able to identify countries in this plot, I think you'll have to come up with another scheme. One idea (not obligatory, just a suggestion): put lineage on the inner annotation ring and then label countries with color and a letter (like the numbering of lineages in supplemental figure 2).

We agree that this is challenging, although the key point here is that the European countries are entirely admixed. To address this point, we have now added additional colours track to SupplementaryFigure 1, showing only the European countries (with a different colour scale), and we have done the same for the North American countries, and the remaining 'other' countries.

Supplementary Figure 4. I believe the title of the middle annotation column is missing a letter. Done, thank you.

Supplementary Figure 6. Quibble: the Nichols reference genomes aren't unrelated, they're in a separate clade.

Done, we have rephrased this to "distinct from".

Supplementary Figure 7. Unclear how to read the plot on the right. Does the black/gray mean the same as in the key to the left? If black is 'direct from clinical sample' what does that mean for, say, lineage 8?

Yes, the colour scheme is consistent between the two plots in Supplementary Figure 7. Sublineage 8 represents a Nichols sublineage containing 15 samples, of which 4 had some passage in the rabbit. We know the 3 "UW" samples from sublineage 8 were passaged in the rabbit for only 2 rounds, but we did not make a distinction here between "extensive passage", which is not always known or quantifiable, and "minimal passage". To make this proportional plot easier to understand, we have now added the exact sample counts to the figure, and clarified the labelling in the legend.

Decision Letter, first revision:

Our ref: NMICROBIOL-21030847A

14th September 2021

Dear Mathew,

Thank you for your patience as we've prepared the guidelines for final submission of your Nature Microbiology manuscript, "Contemporary syphilis is characterised by rapid global spread of pandemic *Treponema pallidum* lineages" (NMICROBIOL-21030847A). Please carefully follow the step-by-step instructions provided in the attached file, and add a response in each row of the table to indicate the changes that you have made. Please also check and comment on any additional marked-up edits we have proposed within the text. Ensuring that each point is addressed will help to ensure that your revised manuscript can be swiftly handed over to our production team.

In recognition of the time and expertise our reviewers provide to Nature Microbiology's editorial process, we would like to formally acknowledge their contribution to the external peer review of your manuscript entitled "Contemporary syphilis is characterised by rapid global spread of pandemic *Treponema pallidum* lineages". For those reviewers who give their assent, we will be publishing their names alongside the published article.

Nature Microbiology offers a Transparent Peer Review option for new original research manuscripts submitted after December 1st, 2019. As part of this initiative, we encourage our authors to support increased transparency into the peer review process by agreeing to have the reviewer comments, author rebuttal letters, and editorial decision letters published as a Supplementary item. When you submit your final files please clearly state in your cover letter whether or not you would like to participate in this initiative. Please note that failure to state your preference will result in delays in accepting your manuscript for publication.

Cover suggestions

As you prepare your final files we encourage you to consider whether you have any images or illustrations that may be appropriate for use on the cover of Nature Microbiology.

Nature Microbiology has now transitioned to a unified Rights Collection system which will allow our Author Services team to quickly and easily collect the rights and permissions required to publish your work. Approximately 10 days after your paper is formally accepted, you will receive an email in providing you with a link to complete the grant of rights. If your paper is eligible for Open Access, our Author Services team will also be in touch regarding any additional information that may be required to arrange payment for your article.

Please note that *Nature Microbiology* is a Transformative Journal (TJ). Authors may publish their research with us through the traditional subscription access route or make their paper immediately open access through payment of an article-processing charge (APC). Authors will not be required to make a final decision about access to their article until it has been accepted. [Find out more about Transformative Journals](https://www.springernature.com/gp/open-research/transformative-journals)

Authors may need to take specific actions to achieve [compliance with funder and institutional open access mandates](https://www.springernature.com/gp/open-research/funding/policy-compliance-faqs) For submissions from January

2021, if your research is supported by a funder that requires immediate open access (e.g. according to [Plan S principles](https://www.springernature.com/gp/open-research/plan-s-compliance)) then you should select the gold OA route, and we will direct you to the compliant route where possible. For authors selecting the subscription publication route our standard licensing terms will need to be accepted, including our [self-archiving policies](https://www.springernature.com/gp/open-research/policies/journal-policies). Those standard licensing terms will supersede any other terms that the author or any third party may assert apply to any version of the manuscript.

Please use the following link for uploading these materials:
[REDACTED]

Best regards,
[REDACTED]

Reviewer #1:

Remarks to the Author:

The authors of the article responded well to my questions, but also to those of the other three reviewers. They also added analyzes and figures which complete and enrich the manuscript without affecting its main message. This article is mature and should have a real impact in terms of scientific interest but also with the general public. I am therefore very much in favor of its publication in Nature Microbiology.

Reviewer #2:

Remarks to the Author:

The authors have dealt with all of my comments.

Reviewer #3:

Remarks to the Author:

The authors have substantially revised the manuscript following almost all the recommendations by myself and the other reviewers. I am pleased to say that the manuscript has improved and that my quibbles are now solved. I recommend this manuscript for publication in Nat. Microb.

Reviewer #4:

Remarks to the Author:

I appreciate the authors' thoughtful and thorough responses and their work on the revision. This will be an important contribution to the field and will spur much interest and help shape further studies.

Final Decision Letter:

Dear Mat,

I am pleased to accept your Article "Global phylogeny of *Treponema pallidum* lineages reveals recent expansion and spread of contemporary syphilis" for publication in Nature Microbiology. Thank you for having chosen to submit your work to us and many congratulations.

Before your manuscript is typeset, we will edit the text to ensure it is intelligible to our wide readership and conforms to house style. We look particularly carefully at the titles of all papers to ensure that they are relatively brief and understandable.

Acceptance of your manuscript is conditional on all authors' agreement with our publication policies (see www.nature.com/nmicrobiolate/authors/gta/content-type/index.html). In particular your manuscript must not be published elsewhere and there must be no announcement of the work to any media outlet until the publication date (the day on which it is uploaded onto our website).

Please note that *Nature Microbiology* is a Transformative Journal (TJ). Authors may publish their research with us through the traditional subscription access route or make their paper immediately open access through payment of an article-processing charge (APC). Authors will not be required to make a final decision about access to their article until it has been accepted. [Find out more](https://www.springernature.com/gp/open-research/transformative-journals)

about Transformative Journals

Authors may need to take specific actions to achieve compliance with funder and institutional open access mandates. For submissions from January 2021, if your research is supported by a funder that requires immediate open access (e.g. according to [Plan S principles](https://www.springernature.com/gp/open-research/plan-s-compliance)) then you should select the gold OA route, and we will direct you to the compliant route where possible. For authors selecting the subscription publication route our standard licensing terms will need to be accepted, including our [self-archiving policies](https://www.springernature.com/gp/open-research/policies/journal-policies). Those standard licensing terms will supersede any other terms that the author or any third party may assert apply to any version of the manuscript.

Congratulations once again and I look forward to seeing the article published.